# Structural basis for the recognition of spliceosomal SmN/B/B' proteins by the RBM5 OCRE domain in splicing regulation

André Mourão[1,2†], Sophie Bonnal[3†], Komal Soni[1,2†], Lisa Warner[1,2†], Rémy Bordonné[4], Juan Valcárcel[3,5*], Michael Sattler[1,2*]

[1]Institute of Structural Biology, Helmholtz Zentrum München, Neuherberg, Germany; [2]Biomolecular NMR and Center for Integrated Protein Science Munich, Department Chemie, Technische Universität München, Garching, Germany; [3]Barcelona Institute of Science and Technology and Universitat Pompeu Fabra, Centre de Regulació Genòmica, Barcelona, Spain; [4]Institut de Génétique Moléculaire de Montpellier, CNRS-UMR5535, Université de Montpellier, Montpellier, France; [5]Institució Catalana de Recerca i Estudis Avançats, Barcelona, Spain

**\*For correspondence:**
juan.valcarcel@crg.es (JVá);
sattler@helmholtz-muenchen.de
(MS)

[†]These authors contributed
equally to this work

**Competing interest:** See
page 22

**Reviewing editor:** Douglas L
Black, University of California,
Los Angeles, United States

**Abstract** The multi-domain splicing factor RBM5 regulates the balance between antagonistic isoforms of the apoptosis-control genes *FAS/CD95*, *Caspase-2* and *AID*. An OCRE (OCtamer REpeat of aromatic residues) domain found in RBM5 is important for alternative splicing regulation and mediates interactions with components of the U4/U6.U5 tri-snRNP. We show that the RBM5 OCRE domain adopts a unique β–sheet fold. NMR and biochemical experiments demonstrate that the OCRE domain directly binds to the proline-rich C-terminal tail of the essential snRNP core proteins SmN/B/B'. The NMR structure of an OCRE-SmN peptide complex reveals a specific recognition of poly-proline helical motifs in SmN/B/B'. Mutation of conserved aromatic residues impairs binding to the Sm proteins *in vitro* and compromises RBM5-mediated alternative splicing regulation of FAS/CD95. Thus, RBM5 OCRE represents a poly-proline recognition domain that mediates critical interactions with the C-terminal tail of the spliceosomal SmN/B/B' proteins in *FAS/CD95* alternative splicing regulation.
DOI: https://doi.org/10.7554/eLife.14707.001

## Introduction

An essential step during the regulation of eukaryotic gene expression is the removal of non-coding intron sequences from pre-mRNA transcripts through the process of pre-mRNA splicing. The catalytic steps of pre-mRNA splicing are carried out by the spliceosome, a large and dynamic assembly of five small nuclear ribonucleoprotein (snRNP) complexes and more than 150 additional splicing factor proteins (*Wahl et al., 2009*). Many splicing factors are involved in early steps of the assembly of the spliceosome through the recognition of short regulatory RNA motifs and/or through protein-protein interactions. Alternative splicing is the mechanism by which particular intronic or exonic regions are included or excluded to produce diverse mRNAs from the same gene (*Blencowe, 2006*). It is thought that more than 90% of human multi-exon genes undergo alternative splicing (*Pan et al., 2008*; *Wang et al., 2008*). The genomic diversity of eukaryotic gene expression is thus greatly expanded by alternative splicing of mRNA transcripts. Often, the protein products of alternative splicing have antagonistic roles in cellular functions and are implicated in human diseases (*Cooper et al., 2009*). Notably, mutations in splicing factors that modulate alternative splicing decisions have been implicated in cancer (*David and Manley, 2010*; *Bonnal et al., 2012*).

**eLife digest** The information required to produce proteins is encoded within genes. In the first step of creating a protein, its gene is "transcribed" to form a pre-messenger RNA molecule (called pre-mRNA for short). Both the gene and the pre-mRNA contain regions called exons that code for protein, and regions called introns that do not. The pre-mRNA therefore undergoes a process called splicing to remove the introns and join the exons together into a final mRNA molecule that is "translated" to make the protein.

Many pre-mRNAs can be spliced in several different ways to include different combinations of exons in the final mRNA molecule. This process of "alternative splicing" allows different versions of a protein to be produced from the same gene. Changes that alter the pattern of alternative splicing in a cell affect various cellular and developmental processes and have been linked to diseases such as cancer.

The pre-mRNA transcribed from a gene called *FAS* can be alternatively spliced so that it either does or does not contain an exon that enables the protein to embed itself in the cell membrane. The protein produced from mRNA that includes this exon generates a cell response that leads to cell death. By contrast, protein produced from mRNA that lacks this exon is released from cells and promotes their survival. A splicing factor called RBM5 promotes the removal of this exon from *FAS* pre-mRNA.

RBM5 binds to some of the proteins that make up the molecular machine that splices pre-mRNA molecules. Mourão, Bonnal, Soni, Warner et al. have now used a technique called nuclear magnetic resonance spectroscopy to solve the three-dimensional structure formed when RBM5 binds to one of these proteins, called SmN. Further experiments introduced specific mutations to the proteins to investigate their effects in human cells. This revealed that mutations that impaired the association between RBM5 and SmN compromised the activity of RBM5 to regulate the alternative splicing of *FAS* pre-mRNA molecules.

Future research could examine how RBM5 associates with pre-mRNAs and other components of the splicing machinery, and investigate whether proteins that are closely related to RBM5 act in similar ways.

DOI: https://doi.org/10.7554/eLife.14707.002

A biologically important example of alternative splicing is found in the *FAS* gene (also known as *CD95* or *APO-1*). The *FAS* gene encodes a transmembrane signaling protein that stimulates a pro-apoptotic signaling cascade upon binding of the FAS ligand at the cell surface (*Krammer, 2000*). Alternatively spliced *FAS* transcripts that exclude exon 6 encode a soluble Fas isoform that lacks the transmembrane domain. This soluble isoform can be secreted outside of the cell where it sequesters the FAS ligand and inhibits downstream activation of apoptosis (*Cheng et al., 1994*; *Cascino et al., 1995*). Thus, regulation of the alternative splicing of *FAS* can either stimulate or inhibit cell survival. The pro-apoptotic Fas protein plays an important role during T-lymphocyte maturation (*Liu et al., 1995*; *Papoff et al., 1996*; *Van Parijs et al., 1998*; *Roesler et al., 2005*) and additional evidence implicates this isoform in the proliferation of cancer cells (*Chen et al., 2010*).

A number of splicing factors have been shown to modulate *FAS* alternative splicing, including RBM5. The multi-domain RNA-binding protein 5 (RBM5) regulates *FAS* splicing by promoting skipping of exon 6 (*Bonnal et al., 2008*). RBM5 is a 92 kDa, multi-domain protein with an arginine-serine (RS)-rich region, two RNA Recognition Motifs (RRM1 and RRM2), two Zinc-Finger domains (ZF1 and ZF2), a C-terminal OCtamer REpeat (OCRE) domain (*Callebaut and Mornon, 2005*) in addition to a glycine patch, and KEKE (lysine/glutamate) repeats (*Figure 1*). It belongs to the family of RNA Binding Motif (RBM) proteins, including RBM6 and RBM10, which share a similar domain organization with RBM5 and have 30% and 50% amino acid identity, respectively (*Sutherland et al., 2005*). The RBM5 (also known as H37 and LUCA-15) and RBM6 genes are located in a chromosomal region 3p21.3, which is frequently deleted in heavy smokers and lung cancers (*Oh et al., 2002*; *Zabarovsky et al., 2002*). RBM5 is known to regulate the alternative splicing of apoptosis–related genes, such as *FAS* and *Caspase-2* (*Bonnal et al., 2008*; *Fushimi et al., 2008*). It has also been reported to suppress metastasis by modulating the expression of Rac1, β-catenin, collagen and

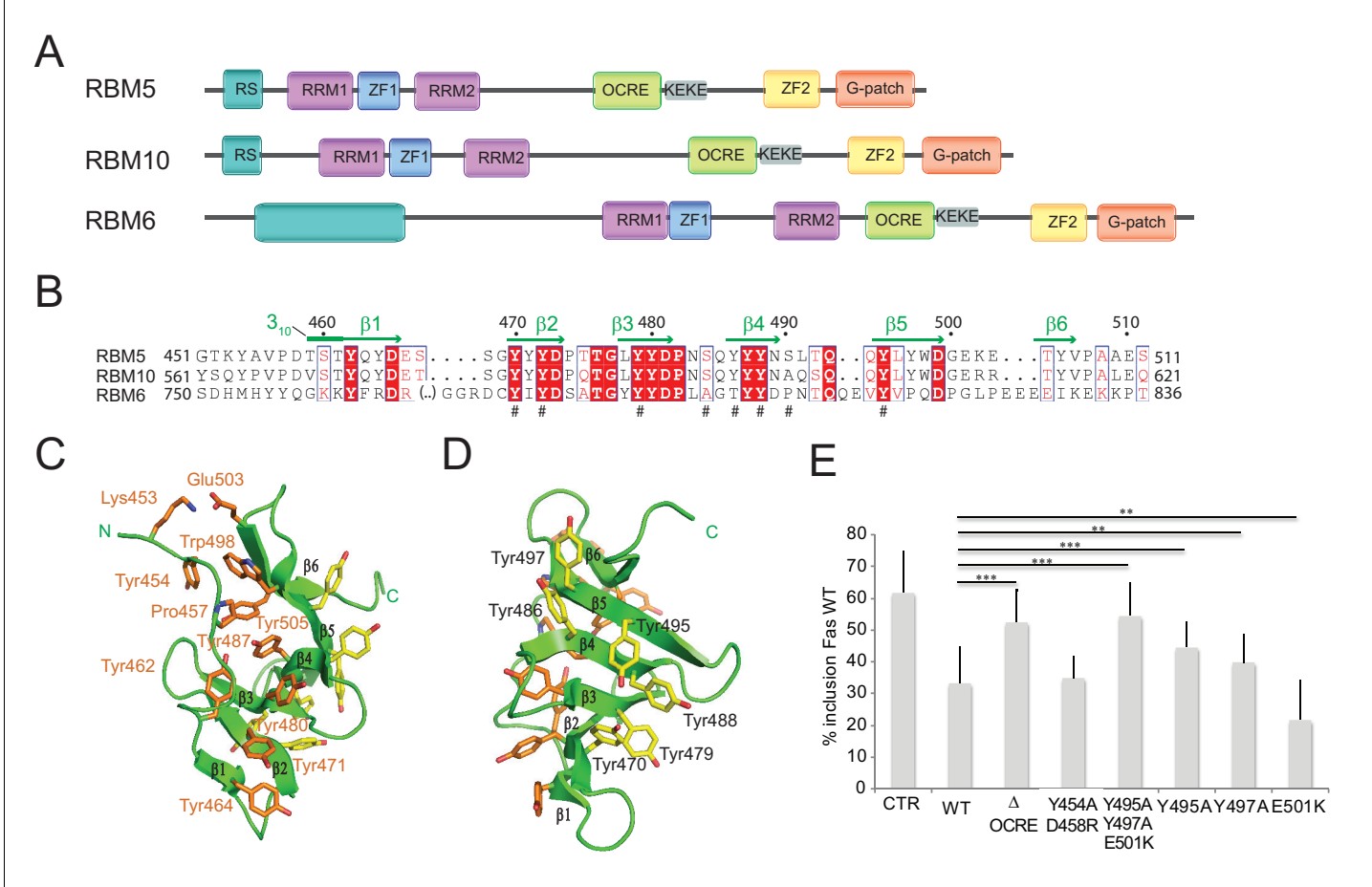

**Figure 1.** Structure and functional analysis of the RBM5 OCRE domain. (**A**) Domain composition of the related RBM5, RBM6 and RBM10 family proteins. (**B**) Sequence alignment of OCRE domains of human RBM5, RBM10 and RBM6 proteins. (**C,D**) Two views in a cartoon presentation of the RBM5 OCRE domain, showing side chains of conserved and exposed tyrosine residues, some of which were probed by mutational analysis. (**C**) At one side of the β-sheet surface the N-terminal extension shields the tyrosines (orange) from solvent exposure, (**D**) on the opposite surface of the β-sheet, numerous tyrosine residues (yellow) are solvent accessible. (**E**) Mutational analysis of conserved residues of the RBM5 OCRE domain. Specific mutations of conserved and accessible residues of RBM5 OCRE domain impair the activity of the protein in *FAS* alternative splicing regulation ex vivo. HeLa cells were co-transfected with a Fas wild type alternative splicing reporter (harbouring sequences between the 5' end of exon 5 and the first 47 nucleotides of exon 7) and T7-RBM5 expression plasmids. RNA and proteins were isolated 24 hr after transfection. Patterns of alternative splicing were studied by RT-PCR using specific primers (PT1, PT2) and the percentage of inclusion was calculated and is presented in the histogram for a minimum of 16 replicas of the experiment. T-test (two-tailed distribution, homoscedastic) results are mentioned (**<0,01; ***<0,001). Full quantification and T-test results are provided in *Figure 5—figure supplement 2*.

DOI: https://doi.org/10.7554/eLife.14707.003

The following figure supplement is available for figure 1:

**Figure supplement 1.** NMR analysis of the RBM5 OCRE domain.

DOI: https://doi.org/10.7554/eLife.14707.004

laminin (*Oh et al., 2010*). However, RBM5 has also been found to be up-regulated in some aggressive forms of breast and ovarian cancers (*Oh et al., 1999*; *Rintala-Maki et al., 2007*). The possible dual role of RBM5 in cancer progression may be linked to its various activities as splicing regulator, including differential recognition of its pre-mRNA targets.

In the regulation of *FAS* alternative splicing, RBM5 inhibits mature spliceosome formation by blocking association of the U4/U6.U5 tri-snRNP complex (in complex B) after the splice sites flanking exon 6 have been recognized by U1 and U2 snRNPs (*Bonnal et al., 2008*). RBM5 therefore appears to influence the pairing between the splice sites after complex A formation and thus promote tri-snRNP assembly at the distal splice sites, leading to exon 6 skipping (*Bonnal et al., 2008*).

Moreover, it was shown that the OCRE domain of RBM5 interacts with components of the tri-snRNP and is essential for the function of RBM5 in regulating *FAS* splicing (*Bonnal et al., 2008*).

OCRE domains, which are present in different proteins in various organisms (*Xiao et al., 2013*), were identified by bioinformatic sequence analysis as a tandem of five imperfect octameric repeat sequences with triplets of aromatic residues (*Callebaut and Mornon, 2005*), expected to predominantly adopt β-strand secondary conformation. Due to the presence of OCRE domain in RNA binding proteins such as RBM5, they were presumed to facilitate RNA binding (*Callebaut and Mornon, 2005*). However, at least in the case of RBM5, the OCRE domain appears to play a role in mediating protein-protein interactions (*Bonnal et al., 2008*).

In order to provide molecular insights into the function of the OCRE domain in splicing regulation, we have combined structural and biochemical experiments with mutational analyses *in vitro* and *in vivo*. We show that the RBM5 OCRE domain directly binds to the spliceosomal SmN/B/B' proteins. Sm proteins are core components of most spliceosomal snRNPs and form a heptameric ring composed of SmD1, SmD2, SmF, SmE, SmG, SmD3 and SmN/B/B', which jointly recognize the uridine-rich Sm site RNA motif in U1, U2, U4 and U5 snRNAs (*Kambach et al., 1999*; *Will and Lührmann, 2001*; *Pomeranz Krummel et al., 2009*; *Weber et al., 2010*; *Leung et al., 2011*; *Matera and Wang, 2014*; *Kondo et al., 2015*). SmB and SmB' are encoded by two transcript variants from the *SNRPB* gene, while a different gene, highly expressed in brain and heart, encodes the homologous protein SmN (*Schmauss et al., 1992*). SmN/B/B', SmD1, and SmD3 have C-terminal extensions that include symmetrically dimethylated RG repeats (*Brahms et al., 2001*; *Tripsianes et al., 2011*). The SmN/B/B' C-terminal tails contain additional proline-rich sequences, where these regions in SmN, SmB and SmB' are 93% homologous (*van Dam et al., 1989*).

Here, we show that the RBM5 OCRE domain binds to the C-terminal proline-rich motifs present in SmB and SmN. The structure of the RBM5 OCRE domain adopts a unique β-sheet fold that recognizes the proline-rich C-terminal tails of the SmN/B/B' proteins through aromatic-CH interactions. We demonstrate that disruption of these interactions by mutations in the OCRE domain or in the proline-rich motifs of its ligands decreases the affinity between SmN/B/B' and RBM5 *in vitro* and affects alternative splicing regulation of the FAS gene. Our results demonstrate that OCRE is a novel protein-protein interaction domain that mediates interactions with the core spliceosome in alternative splicing regulation.

## Results

### The RBM5 OCRE domain adopts a unique β-sheet fold involved in splicing regulation

To gain insights into the molecular functions of the RBM5 OCRE domain, we determined the three-dimensional structure for the human RBM5 OCRE domain using solution NMR techniques. Previous reports (*Callebaut and Mornon, 2005*) and a multiple sequence alignment of the related splicing factors RBM5, RBM6, and RBM10 indicated that the OCRE domain spans amino acid residues 451–511 of RBM5 (*Figure 1A,B*). NMR chemical shift and $^{15}$N relaxation analyses of a construct with a C-terminal extension (451–532, including the KEKE region) did not reveal additional structural elements, suggesting that residues 451–511 indeed define the OCRE fold (*Figure 1—figure supplement 1A,B*). An analysis of $^{13}$C secondary chemical shifts shows that the secondary structure of the RBM5 OCRE domain comprises six β-strands between residues 460 and 505 (*Figure 1B*; *Figure 1—figure supplement 1C*). Notably, residues 452–459 preceding β1, and the loops connecting the β-strands are well-structured and not flexible (*Figure 1—figure supplement 1B,C*).

The solution structure of the RBM5 OCRE domain is shown in *Figures 1C,D* and *Figure 1—figure supplement 1D,E*; structural statistics are provided in *Table 1*. Residues 462–465 (β1), 470–473 (β2), 478–481 (β3), 486–489 (β4), 494–498 (β5), and 504–506 (β6) make up six consecutive anti–parallel β-strands. An N-terminal extension (residues 452–459) packs against one side of the β–sheet and is connected to the β1 strand by a $3_{10}$ helix (residues 459–461) (*Figures 1C,D*). The six β–strands form a twisted β–sheet where aromatic side chains are exposed on opposite surfaces of the β–sheet (*Figure 1C,D*). On one side, the aromatic side chains of Tyr464 (in strand β1), Tyr 471 (β2), Tyr480 (β3), Tyr487 (β4) form extended aromatic side chain interactions on the surface. Tyr487 forms an additional hydrophobic cluster with Tyr462 and Pro457 from the N-terminal extension. Trp498 and

**Table 1.** Structural statistics RBM5 OCRE and OCRE/SmN complex.

| | OCRE | OCRE + SmN |
|---|---|---|
| **NMR distance and dihedral restraints** | | |
| *Distance restraints* | | |
| Total NOE | 1171 | 1127 |
| Intra-residue | 220 | 419 |
| Inter-residue | | |
| Sequential ($|i-j| = 1$) | 292 | 118 |
| Medium-range ($|i-j| < 4$) | 156 | 114 |
| Long-range ($|i-j > 5$) | 503 | 329 |
| Hydrogen bonds | 20 | 11 |
| Protein-peptide intermolecular | N/A | 109 |
| *Dihedral angle restraints* | | |
| φ | 47 | 51 |
| ψ | 51 | 51 |
| **Structure statistics** | | |
| *Violations (mean and s.d.)* | | |
| Distance restraints (>0.2 Å) | 0.685 ± 0.363 | 0.324 ± 0.111 |
| Dihedral angle restraints (>5 °) | 0 ± 0 | 0 ± 0 |
| Max. distance restraint violation (Å) | 0.657 ± 0.470 | 0.587 ± 0.142 |
| Max. dihedral angle restraint violation (°) | 0 | 0 |
| *Deviations from idealized geometry* | | |
| Bond lengths (Å) | 0.0038 ± 0.0002 | 0.0046 ± 0.0001 |
| Bond angles (°) | 0.444 ± 0.026 | 0.616 ± 0.015 |
| Impropers (°) | 1.230 ± 0.0681 | 1.737 ± 0.00889 |
| *Average pairwise r.m.s.d.\* (Å)* | | |
| Heavy | 0.72 ± 0.05 | 0.71 ± 0.10 |
| Backbone | 0.38 ± 0.07 | 0.38 ± 0.09 |

\*Pairwise r.m.s.d. was calculated among 10 refined structures for residues 455-508 (RBM5 OCRE) and 221-229 (SmN) after water refinement.

DOI: https://doi.org/10.7554/eLife.14707.005

Tyr505 from the C-terminal extension, and the N-terminal Tyr454 are also involved in this cluster (*Figure 1C*). This forms a compact fold that is further stabilized by a salt bridge involving Lys453 and Glu503 and thus brings the N- and C-terminal regions of the OCRE domain in close spatial proximity. Due to the interactions with residues from the N-terminal extension this aromatic surface of the OCRE domain is shielded from the solvent. In contrast, on the other side of the β-sheet, the interactions of the aromatic side chains of Tyr470 (β2), Tyr479 (β3), Tyr486, Tyr488 (β4), and Tyr495, Tyr497 (β5) form an aromatic surface with the tyrosine hydroxyl groups exposed to the solvent (*Figure 1D*). The N-terminal extension has an extended conformation consistent with secondary $^{13}$C chemical shifts (*Figure 1—figure supplement 1C*). Although the topology of the six antiparallel β-strands is rather simple, the twisted β-sheet of the OCRE domain appears unique. Structural similarity searches with Dali (*Holm et al., 2008*) and SSM (*Krissinel and Henrick, 2004*) did not reveal any significant structural homologs (Z-scores <2), indicating that the OCRE domain represents a unique fold. While the β-sheet fold is quite simple, the specific arrangement of the N-terminal extension represent the unique features of the OCRE fold. The electrostatic surface potential of the OCRE domain is predominantly negatively charged with some hydrophobic patches (*Figure 1—figure supplement 1E*), consistent with a potential role in protein–protein interactions and less likely for nucleic acid binding.

To get initial insights into functionally relevant residues and subdomains, we focused on surface-exposed amino acids conserved between OCRE domains from different RBM proteins. These include Tyr495, Tyr497 on one surface of the fold and Tyr454, Glu501 and Asp458 on the opposite side. To assess the functional relevance of these residues, we generated triple and double point mutations (Y495/Y497/E501(YYE)>AAK) and (Y454/458(YD)>AR), respectively. The mutants were expressed in HeLa cells, co-transfected with a *FAS* alternative splicing reporter (*Forch et al., 2000*) and the pattern of Fas exon 6 inclusion/skipping analyzed by RT-PCR. While the YD mutant had similar activity as the WT protein in promoting Fas exon 6 skipping, the YYE mutation failed to induce significant levels of *FAS* exon 6 skipping, similar to the effect of deleting the entire OCRE domain (*Figure 1E*). To further investigate the importance of the residues in the YYE cluster, single point mutants were generated (Y495A, Y497A, E501D/K) and tested by co-transfection. Replacement of the aromatic tyrosines that are exposed on one surface of the β-sheet to alanine impaired the function of the protein in FAS alternative splicing regulation, arguing that the tyrosine residues in the surface of the β-sheet are important for the function of the OCRE domain in splicing regulation (*Figure 1E*).

## RBM5 OCRE domain binds to the C-terminal tails of SmN/B/B'

It was previously shown that the RBM5 OCRE domain mediates interactions with protein components of the U5 snRNP complex, but the direct binding partners were not determined (*Bonnal et al., 2008*). SmN, a core snRNP protein highly homologous to SmB/B', was among proteins, identified by mass-spectrometry, pulled down with GST-OCRE RBM5 from HeLa cell extracts (*Bonnal et al., 2008*). We also found RBM5 as an interacting partner in a two-hybrid screen using the C-terminal tail of human SmB as bait (unpublished results). SmN/B/B' are components of the Sm core, a heptamer of Sm proteins that associates with all spliceosomal snRNPs except U6. While SmB and SmB' are ubiquitous, SmN is primarily found in neuronal and cardiac cells but has also been found to be expressed in HeLa cells (*Sharpe et al., 1990*). The three SmN/B/B' isoforms have a similar domain architecture, comprising a small globular N-terminal Sm domain (*Kambach et al., 1999*) and a C-terminal region that is expected to be intrinsically disordered. The C-terminal region starts with an arginine-glycine (RG)-rich region at ~residue 90 that harbors symmetrically dimethylated arginine residues, and additionally comprises a 60–70 residue proline-rich region of unknown function beyond residue 167 (*Figure 2A*).

As the globular N-terminal Sm domain in SmN/B/B' proteins participates in the formation of the heptameric Sm core in the U snRNP complexes (*Kambach et al., 1999*; *Pomeranz Krummel et al., 2009*) it seems unlikely that it could interact with the RBM5 OCRE domain. We thus considered the possibility that, in particular, the C-terminal domain of SmN/B/B' could directly bind to the RBM5 OCRE domain. As an initial test, we verified that recombinant purified GST-RBM5, or a derivative containing the C-terminal (OCRE-containing) domain, pulled down *in vitro* translated SmB, while a GST-fusion of the N-terminal domain of RBM5 did not (*Figure 2B*). Next we carried out pull-down experiments using recombinant GST-tagged RBM5 OCRE domains (wild type and mutants where surface-exposed tyrosine residues are replaced by alanine) and assessed the ability to pull down purified recombinant N-terminal T7 epitope-tagged-SmN protein produced in mammalian cells (*Figure 2C*; *Figure 2—figure supplement 1A*). These experiments demonstrate that recombinant GST-tagged OCRE domain binds SmN *in vitro* and that the mutations Y479A, Y488A, or Y495A diminish SmN binding (*Figure 2C*), consistent with the decrease in activity of Y495A in splicing regulation (*Figure 1E*). The mutations Y486A, Y497A and E501K do not significantly impair the interaction (*Figure 2C*). Interestingly, mutation of Y495 to tryptophan or phenylalanine did not compromise binding, arguing that an aromatic residue at this position is required for the interaction (*Figure 2—figure supplement 1*).

To further characterize these interactions by an independent method, we performed yeast-two-hybrid assays and tested whether the C-terminal tail of SmB binds to different variants of RBM5. These experiments demonstrate a robust interaction between RBM5 and the SmB tail, where the OCRE domain is found to be necessary and sufficient for the interaction (*Figure 2D*). Moreover, mutations of surface-exposed tyrosines to alanine reduce the interaction, while mutation to another aromatic amino acid (phenylalanine) maintains the interaction (*Figure 2E*) as was observed for the full-length SmN protein (*Figure 2C*; *Figure 2—figure supplement 1*).

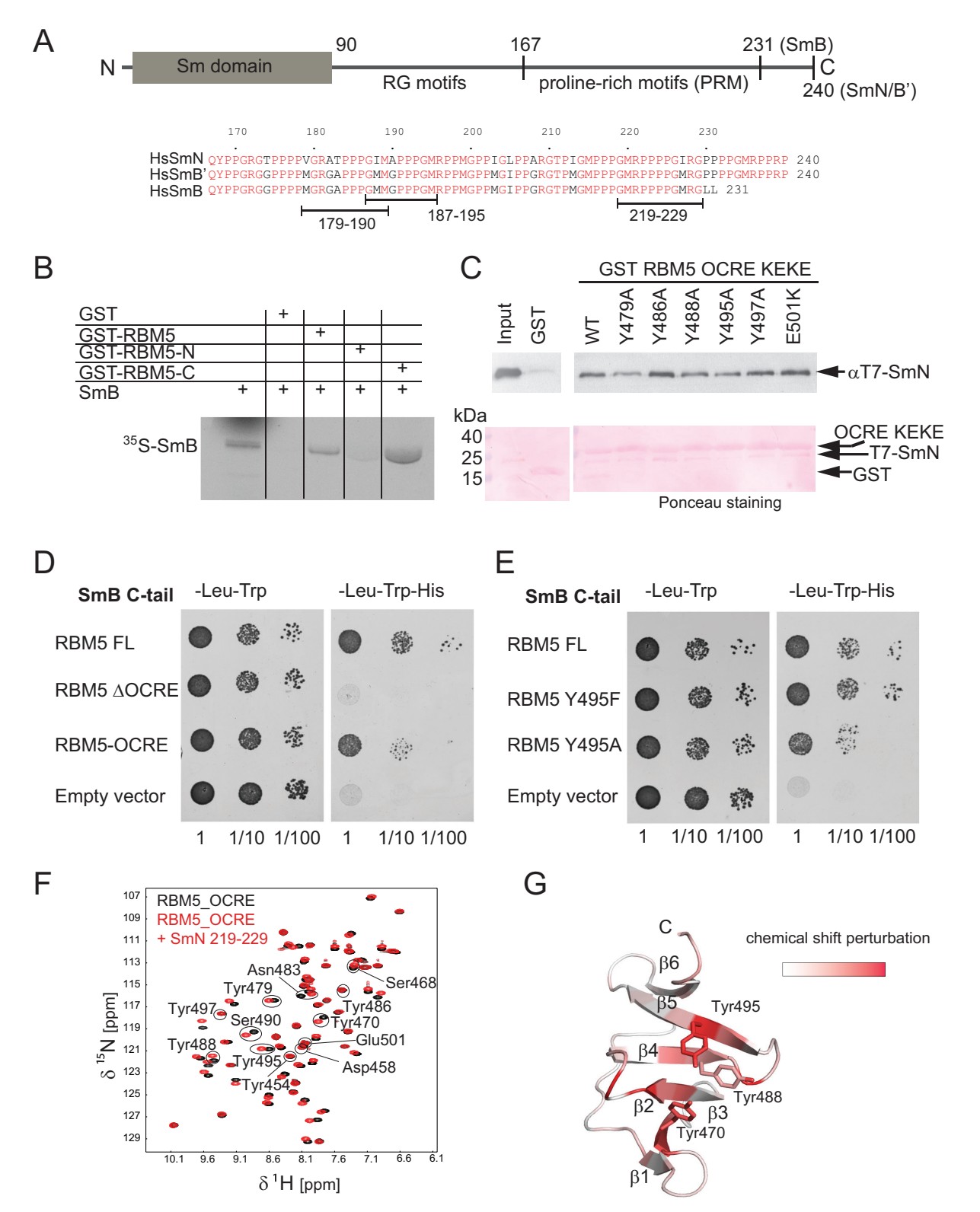

**Figure 2.** Interaction of RBM5 OCRE with SmN/B/B' proteins. (**A**) Domain structure and multiple sequence alignment of the C-terminal tails of human SmN/B/B' proteins. (**B**) *In vitro* pull down assays of $^{35}$S-labeled SmB with GST-RBM5 full-length, N- and C-terminal halves of human RBM5. First lane represents 20% of the input used in the pull down. (**C**) Conserved tyrosine residues in the RBM5 OCRE domain are important for the interaction with SmN protein. GST-pull down assays were carried out with different variants of RBM5 comprising the OCRE and KEKE domains harboring the mutations

*Figure 2 continued on next page*

*Figure 2 continued*

indicated. The detection of the protein was carried out by western blot with T7 epitope antibody and Ponceau staining was performed as a loading control. Quantification is provided in *Figure 2—figure supplement 1*. (D) The C-terminal domain of human SmB binds to the OCRE domain of RBM5 *in vivo*. Yeast two hybrid plasmids encoding the C-terminal domain of human SmB (SmB C-tail) and the indicated RBM5 coding regions were transformed into yeast. Serial dilutions of equivalent amounts of exponentially growing yeast were plated on double and triple dropout media. Growth on -Leu -Trp -His is indicative of an interaction between the tested proteins: RBM5-FL: full length RBM5 protein; RBM5 ΔOCRE: deletion of the RBM5 OCRE domain (amino-acids 452–535); RBM5 OCRE: RBM5 OCRE domain (amino-acids 452–535). (E) Surface-exposed tyrosine residues are important for SmB C-tail binding *in vivo*. RBM5-FL: full length RBM5 protein; RBM5 Y495F: RBM5-FL carrying a tyrosine to phenylalanine substitution at position 495; RBM5-FL Y495A: RBM5 carrying a tyrosine to alanine substitution at position 495. The two-hybrid assay was performed as described in panel d. (F) NMR titrationof $^{15}$N-labeled RMB5 OCRE domain (0.2 mM- black) with SmN residues 219–229 (red) at two-fold molar excess. (G) Mapping of NMR chemical shift perturbations upon titration of the OCRE domain with SmN (residues 219–229) onto a surface representation of the RBM5 OCRE domain structure.

DOI: https://doi.org/10.7554/eLife.14707.006

The following figure supplements are available for figure 2:

**Figure supplement 1.** Aromatic residues in RBM5 OCRE are important for interaction with SmN.

DOI: https://doi.org/10.7554/eLife.14707.007

**Figure supplement 2.** Interaction of RBM5 OCRE with C-terminal regions of SmN and SmB.

DOI: https://doi.org/10.7554/eLife.14707.008

## RBM5 OCRE domain recognizes proline-rich motifs in the SmN/B/B' tails

We next wished to identify the region in the Sm tails that mediates the OCRE interaction by using NMR and isothermal titration calorimetry (ITC) experiments. We studied binding of different constructs of the C-terminal tail of SmN, which harbors a region comprising Arg-Gly motifs (residues 95–134, not shown) and a proline rich region (residues 167–240, up to the C-terminal end, *Figure 2A*) (*Weber et al., 2010*). The interaction of the RBM5 OCRE domain was assessed by monitoring NMR $^{1}$H, $^{15}$N chemical shifts of $^{15}$N-labeled OCRE domain during a titration with different SmN-derived peptides (*Figure 2—figure supplement 2*). Titrations with two different SmN fragments (residues 97–196 and 167–196) that both comprise parts of the proline-rich region gave rise to significant and comparable chemical shift perturbations (CSPs) (*Figure 2—figure supplement 2A,B*), suggesting that it is the proline-rich region, which is common to both peptides, that mediates the interaction. A peptide comprising the complete proline-rich region (residues 167–240) shows comparable chemical shift changes. SmN (167–196) and (167–240), both of which comprise about one half and the complete proline-rich region, bind to RBM5 OCRE with dissociation

**Table 2.** Isothermal titration calorimetry data for the OCRE/SmN/B/B' interaction.

| RBM5_OCRE | WT | Y495A | Y495T | Y495F | Y495W | Y488A | Y486A | Y479A | Y454A/D458K | E501K |
|---|---|---|---|---|---|---|---|---|---|---|
| SmN_167-240 | WT | WT | WT | WT | WT | WT | WT | WT | WT | |
| $K_D$ (µM) | 41 ± 2 | 172 ± 6 | 186 ± 12 | 87 ± 3 | 35 ± 3 | 220 ± 11 | 145 ± 5 | 111 ± 5 | 106 ± 15 | 48 ± 2 |

$K_D$ values were determined from replicate measurements, with standard deviations as indicated.

| RBM5_OCRE | WT | WT | WT | WT | WT | WT |
|---|---|---|---|---|---|---|
| SmN_167-240 | 4P→4A | 4P/3P→4A/3A | 4P/3P→4G/3G | R1→E1 | R2→E2 | R1/R2→E1/E2 |
| $K_D$ (µM) | 29 ± 3 | 103 ± 16 | 192 ± 66 | 66.5 ± 6 | 74 ± 4 | 149.5 ± 46 |

| RBM5_OCRE | WT |
|---|---|
| SmN_167-196 | WT |
| $K_D$ (µM) | 195 ± 21 |

| RBM5_OCRE | WT | | WT | WT |
|---|---|---|---|---|
| SmB_167-231 | WT | | 4P→4A | R1/R2→E1/E2 |
| $K_D$ (µM) | 21 ± 3 | | 20 ± 2 | 159 ± 8 |

DOI: https://doi.org/10.7554/eLife.14707.009

constants of $K_D$ = 195 μM and 41 μM, respectively, as determined by ITC (*Figure 2—figure supplement 2F*; *Table 2*).

SmB and SmN share a poly-proline-rich region in their C-terminal tails, with 80% sequence identity and 98% sequence similarity (*Figure 2A*). Upon titration of the SmB C-terminal region (residues 167–231) to $^{15}$N-labeled OCRE, significant NMR CSPs are observed (*Figure 2—figure supplement 2E*). Moreover, the observed CSPs are very similar to those seen with the SmN titration and affect the same residues, indicating that SmB and SmN bind to the same site in the RBM5 OCRE domain (*Figure 2—figure supplement 2G*). The binding affinity of the SmB and SmN peptides to OCRE, determined by isothermal titration calorimetry (ITC) gave similar $K_D$ values of 21 μM and 41 μM, respectively (*Table 2*). Considering the high sequence similarity between SmB, SmB' and SmN, it is expected that OCRE will interact with proline-rich sequences present in SmB and SmB' as well.

It is interesting to note that the NMR CSPs observed for the OCRE domain when titrated with a peptide comprising just one proline-rich motif (PRM, residues 219–229) (*Figure 2F*) or the larger fragments of SmN are very similar (*Figure 2—figure supplement 2A–D*). This indicates that all SmN fragments harboring PRMs utilize the same binding site on the OCRE domain. The strongest CSPs are observed for the amides of Y470, Y488, and Y495, which map to the cluster of exposed aromatic tyrosines on β2, β4, and β5 of the OCRE domain, respectively (*Figure 2G*). However, the short peptide containing four consecutive proline residues induces comparable CSPs only at much higher peptide:OCRE ratio compared to the longer Sm tail peptides, thus indicating higher binding affinity for the longer Sm tail constructs (*Figure 2—figure supplement 2*). Nearly identical spectral changes are induced by the peptide, suggesting that all PRMs contribute to the overall affinity for the OCRE domain. Thus, the interaction with multiple motifs is enhanced by avidity, e.g., interaction of multiple PRMs within the Sm tails provides increased local concentration of PRM ligand motifs and thereby contributes to the significantly higher affinity of Sm-tails compared to a single PRM.

In order to assess the sequence requirements of PRMs for binding to the OCRE domain, we compared the amino acid sequences of the SmN/B tails (*Figure 3A*). This analysis reveals the presence of multiple (five to six) PRMs harboring three or four consecutive prolines in SmB or SmN, which are flanked by conserved arginine residues within ±3 residues on either side of the poly-proline motif (*Weber et al., 2010*). The role of these conserved sequence features was probed by compared binding affinities of wild type and variant Sm tails to the RBM5 OCRE domain using ITC experiments (*Figure 3A*). Reducing the number or PRMs significantly decreases the binding affinity, consistent with avidity effects and contribution to the overall affinity by the presence of multiple PRMs. Replacing four prolines by four alanines has little effect. This observation is consistent with the fact that consecutive stretches of alanines adopt a poly-proline type II (PPII) helical conformation, as confirmed by CD spectra (*Figure 3—figure supplement 1*). As the SmN/B/B' PRMs adopt a PPII helical conformation when bound to the OCRE domain (see below), the replacement by four alanines can thus be tolerated to some extent. Notably, charge reversal of two arginines flanking the poly-proline motif on either side significantly decreases the binding affinity. These data suggest that a PPII helical conformation flanked by positively charged arginine residues is specifically recognized by the OCRE domain.

The contribution of individual residues within the PRM motif was determined by comparing relative binding affinities of wild type and mutant 11-mer peptides that exhibit sequence features observed in the Sm tails (*Figure 3B*). As the binding affinities of these peptides are beyond the detection limit of ITC, we resorted to a semi-quantitative NMR CSP-based approach. We devised a normalized CSP score for a set of seven amide signals surrounding the binding pocket showing significant CSPs in the OCRE domain to obtain a proxy for the relative binding affinities (see Materials and methods for details) (*Figure 3B—figure supplement 2*). As the comparison between different peptides is based on the CSP score derived from the same set of residues, it indirectly reflects their relative binding affinities. Several themes emerge from this comparison suggesting that the OCRE domain has a preferred binding motif, but can also accommodate some sequence variants. First, wild type peptides with three prolines induce smaller CSPs than those with four proline motifs and binding saturation is achieved only at higher peptide concentration, indicating a lower affinity. PRMs harboring only three consecutive proline residues show 2–3-fold reduced affinity relative to four proline motifs. Second, an APAP motif, which disrupts PPII conformations, has a strongly reduced CSP score, suggesting that the structure and composition of the PPII are important for interaction. Third, both arginines flanking the PRM motif are important for the interaction, as a

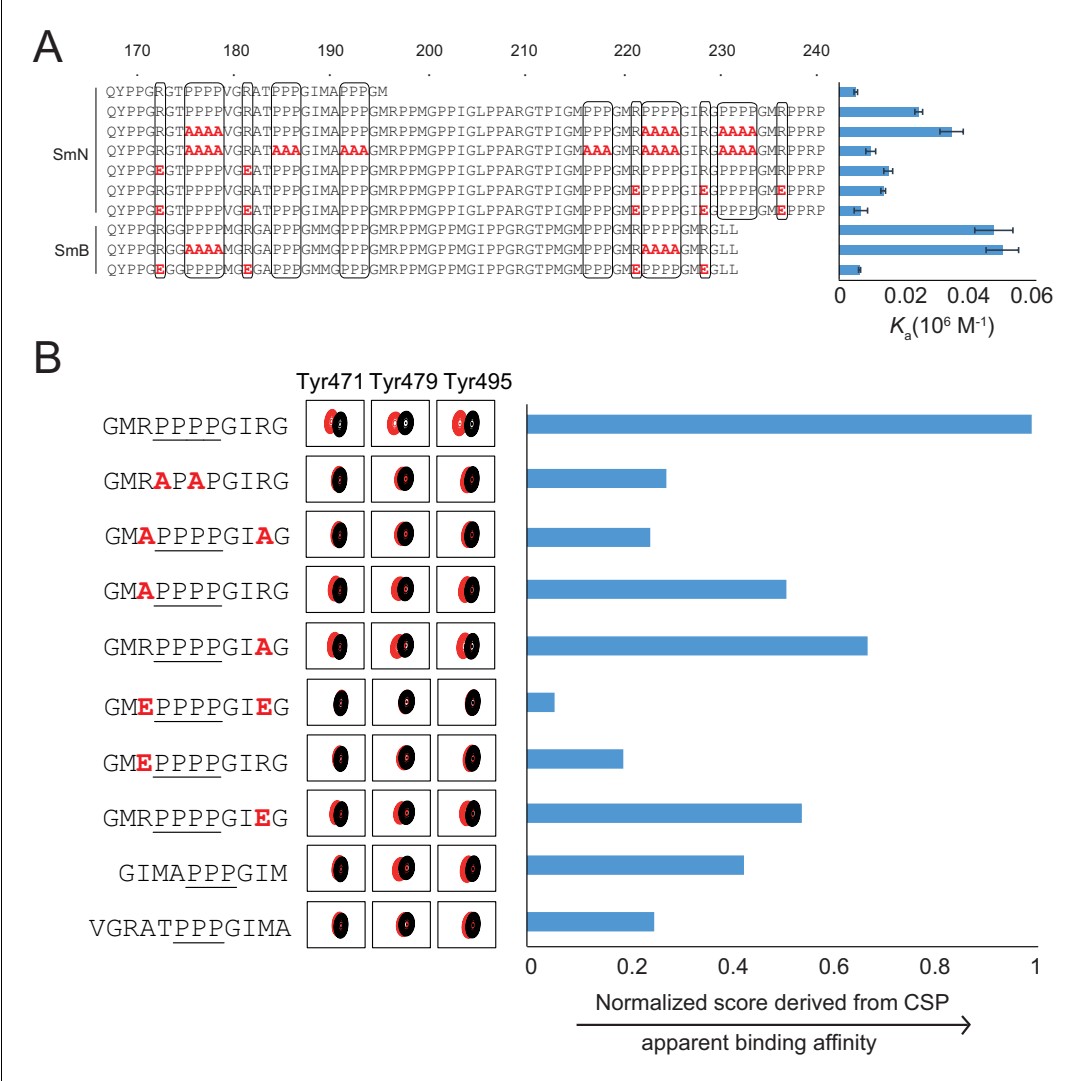

**Figure 3.** Proline-rich motif sequence requirements for OCRE binding. (**A**) Sequence of the C-terminal tail of human SmN/B/B' and mutations of proline-rich-motifs (PRMs) tested. Conserved three or four-proline motifs and flanking arginine residues are highlighted. The binding affinities of these tails to the RBM5 OCRE domain determined by ITC (*Table 2*) are indicated on the right. Note, that association constants are shown here. (**B**) Binding of various PRM peptides to RBM5 OCRE monitored by NMR titrations. Relative binding affinities of PRM peptides to RBM5 OCRE based on a normalized chemical shift perturbation score are shown.

DOI: https://doi.org/10.7554/eLife.14707.010

The following figure supplements are available for figure 3:

**Figure supplement 1.** Circular dichroism spectra of peptides used in the OCRE binding study.
DOI: https://doi.org/10.7554/eLife.14707.011
**Figure supplement 2.** Residues used for normalized CSP score calculation.
DOI: https://doi.org/10.7554/eLife.14707.012

double charge reversal (R→E) strongly reduces the binding affinity. Neutralizing the charge with R→A mutations had a smaller effect, presumably because no charge clashes are introduced with the negatively charged surface of the OCRE domain (*Figure 1—figure supplement 1E*) as is the case for R→E mutations. Taken together, the analysis shown in *Figure 3* indicates that a PPII conformation mediated by four consecutive proline residues with flanking positively charged residues is optimal for OCRE binding. Notably, the arginine residue preceding the PRM appears more important for the interaction than the one located C-terminal to the PRM.

## Solution structure of an RBM5 OCRE/SmN peptide complex

To determine molecular details of the recognition of SmN/B/B' by the RBM5 OCRE domain, we determined its structure in complex with a proline-rich peptide derived from the C-terminal region of SmN (GMRPPPPGIRG) corresponding to SmN residues 219–229 (*Figure 4*; *Figure 4—figure supplement 1A*). This motif is also found within residues 219–229 in SmB with only one amino acid difference (GMRPPPPGMRG). The structure of the complex is defined by numerous NOEs, and is supported by 109 distance restraints derived from intermolecular NOEs (*Figure 4—figure supplement 1B*, *Table 2*). The structure shows that the central proline stretch (SmN Pro222-Pro225) of the bound peptide adopts a PPII helix (*Figure 4*). The PPII conformation is also indicated by the $^{13}$C chemical shifts of the peptide. Considering the large excess of peptide, the chemical shifts largely reflect the unbound state. This indicates that the PPII conformation is already preformed in the

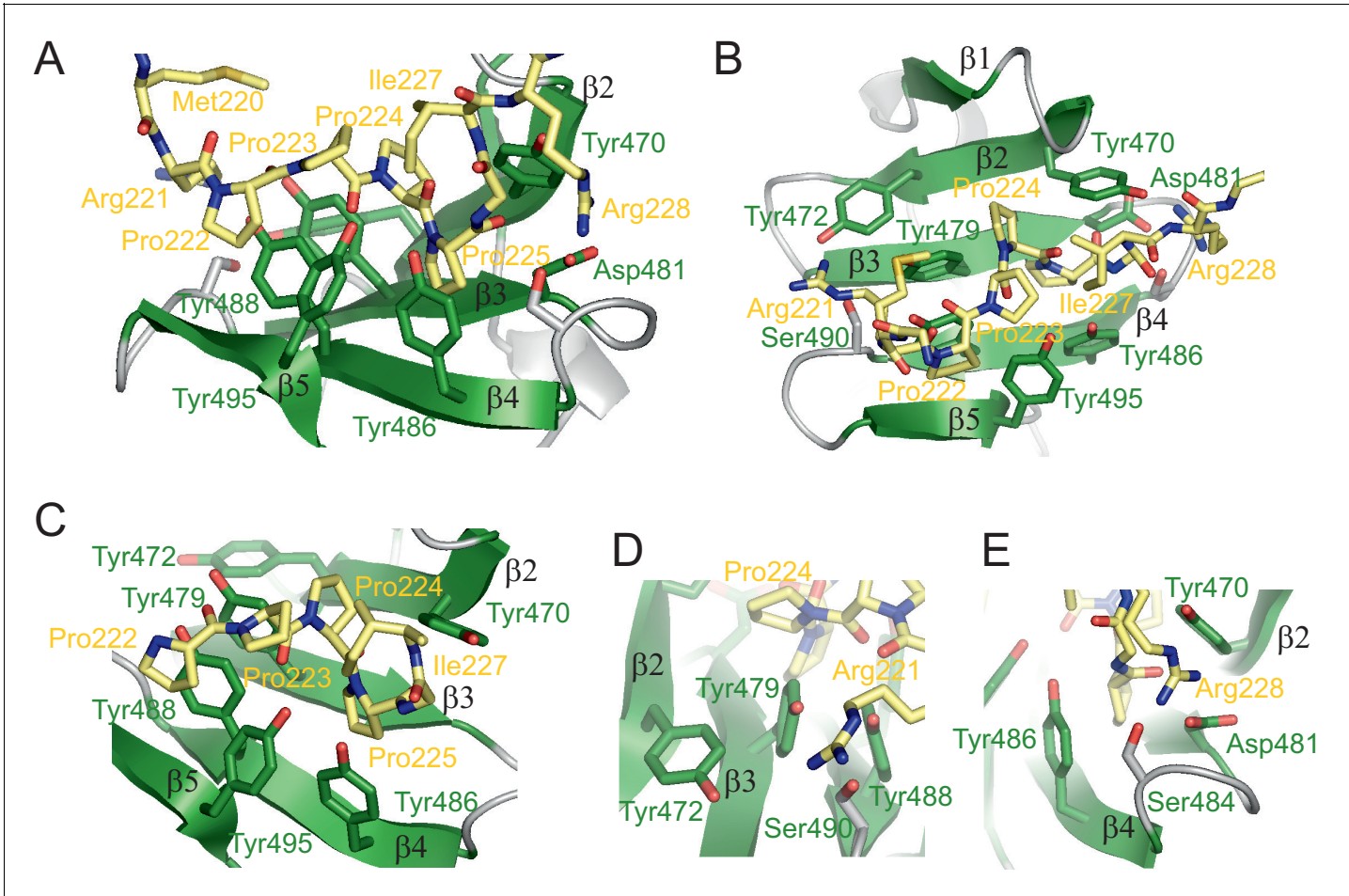

**Figure 4.** Structure of the RBM5 OCRE/SmN peptide complex. (A,B) Side and top views in a cartoon presentation of the RBM5 OCRE/SmN peptide complex. The secondary structure and loops in the RBM5 OCRE domain are colored in green and grey, respectively, the proline-rich motif (PRM) peptide corresponding to SmN residues 221–229 is shown in yellow. (C–E) Zoomed views of key interaction sites. (C) The SmN PRM adopts a proline-type II helical conformation and is recognized by stacking with key tyrosine residues from the OCRE domain. The side chain of SmN Ile227 packs against the hydrophobic surface of the PPII helix. (D) Recognition of SmN Arg221 by interactions with hydroxyl groups of Tyr472, Tyr479 and Ser490 (E) Arg228 forms electrostatic contacts with the side chains of Asp481 and Ser484.

DOI: https://doi.org/10.7554/eLife.14707.013

The following figure supplements are available for figure 4:

**Figure supplement 1.** Structural analysis of the OCRE/SmN complex.
DOI: https://doi.org/10.7554/eLife.14707.014

**Figure supplement 2.** NMR spectra and ITC data for OCRE domain mutants.
DOI: https://doi.org/10.7554/eLife.14707.015

absence of the OCRE domain, consistent with our CD data (*Figure 3—figure supplement 1*). The central part of the proline-rich peptide interacts with conserved tyrosine residues in the OCRE domain in strands β3 (Tyr470, Tyr479), β4 (Tyr486, Tyr488) and β5 (Tyr495) (*Figure 4A–C*). On one side, Tyr488 stacks with SmN Pro222 and on the other with Tyr479, which itself contacts SmN Pro224 by T-stacking. The Tyr486 side chain stacks with SmN Pro225 and interacts with Tyr495, which in turn contacts Pro223 in the SmN peptide.

A unique binding orientation of the peptide is defined by a combination of the stacking interactions with the PPII conformation of the peptide, and interactions with the arginine residues flanking the proline-rich stretch on either side, as well as SmN Ile227. The side chain of SmN Arg221, preceding the poly-proline stretch is poised to interact with the hydroxyl groups of OCRE Tyr472, Tyr479 and Ser490 in loop β1-β2 (*Figure 4D*). The side chain of SmN Arg228 can form potential hydrogen bonds with the side chains of Asp481 and Ser484 in the OCRE domain (*Figure 4E*). Both residues are conserved in loop β3-β4 in the OCRE domains of RBM5 and RBM10 (*Figure 1B*). In addition, the hydrophobic side chain of SmN Ile227, located C-terminal of the proline-rich motif packs against SmN Pro223, consistent with numerous intermolecular NOEs between the two side chains, and thus shields the hydrophobic PPII helix from solvent exposure (*Figure 4C*). To enable these contacts involving the Ile227 side chain, the presence of a preceding Gly226, enables a specific backbone conformation by allowing unusual backbone torsion angles.

Considering the specific spacing of the arginine residues to the four-proline-stretch in the SmN ligand, these interactions define a unique, unambiguous orientation of the peptide along the OCRE β-sheet surface. The structure of the OCRE domain does not undergo significant conformational changes upon binding to the SmN peptide, with a backbone coordinate r.m.s.d. of 1.7Å between the free and SmN-bound OCRE domain. However, the tyrosine side chains involved in the SmN interaction slightly rearrange to optimize contacts with the proline-rich SmN ligand.

To summarize, the RBM5 OCRE domain uses an array of tyrosine side chains that are exposed from one side of its β-sheet to recognize a proline-rich motif in a PPII conformation. Additional specific interactions are mediated by two positively charged side chains flanking the N- and C-terminal sides of the poly-proline helix in the bound SmN peptide, which in part involve the side chain hydroxyl groups of the exposed tyrosines, thus consistent with the strong conservation of tyrosines in the OCRE domain. These structural insights reveal that the OCRE domain represents a novel proline-rich motif binding domain.

## Structure-based mutation analysis *in vitro* and *in vivo*

To evaluate the importance of contacts observed in the OCRE-SmN structure we carried out a mutational analysis of the OCRE domain. We compared the binding of wild-type and mutant OCRE domains to SmN/B/B′ *in vitro* using GST pull-downs, ITC and NMR titrations, and analyzed the functional activity of wild-type and OCRE domain mutants in alternative splicing regulation. We focused on the effects of mutation of tyrosine residues located within the proline-rich motif binding pocket, including Tyr470, Tyr479, Tyr486, Tyr488 and Tyr495 and control mutations of residues that are not involved in the SmN interactions, i.e. Tyr497, Glu501 and Tyr454/Asp458.

To analyze whether the effects of the OCRE mutations could induce (partial) disruption of the OCRE structure, recombinant OCRE proteins were expressed and purified and structural integrity analyzed by NMR (*Figure 4—figure supplement 2*). The NMR spectra of the Y495F, Y495W and E501K OCRE domains suggest that these proteins are globular folded. Small changes in position and intensities of the NMR signals for the Y495F/W mutations likely reflect that aromatic side chains have significant contributions to the chemical shifts of surrounding residues, but are still consistent with the integrity of the overall fold. NMR spectra of the Y495A, Y495T, Y497A and the double mutant Y454A/D458K suggest partial unfolding of strand β5 and the N-terminal extension, respectively, even though the overall fold appears to remain intact. Larger changes observed for the Y479A, Y486A or Y488A OCRE domains suggest more significant destabilization of the fold (*Figure 4—figure supplement 2*).

GST-pulldown experiments (*Figure 2—figure supplement 1*) performed with Tyr495 mutations in the RBM5 OCRE domain confirm the importance of the aromatic side chain of Tyr495 for Sm binding. Next, we compared the binding affinities of wild type OCRE domains with Y495A, Y495F, Y488A and of the double mutant Y454A/D458K using ITC (*Table 2*). Consistent with the structural analysis and the observed effects in splicing regulation, the Y495A and Y488A mutants show strongly

reduced binding to the SmN-derived proline-rich ligands, $K_D$ = 172 and 220 μM, respectively, compared to the wild type OCRE domain ($K_D$ = 41 μM). The Y495F mutation does not strongly affect the SmN interaction $K_D$ = 87 μM. The double mutant Y454A/D458K shows a slightly reduced affinity to SmN $K_D$ = 106 μM, which, however, might reflect partial destabilization of the mutant OCRE fold (*Figure 4—figure supplement 2*). Yeast two-hybrid assays confirmed the importance of an aromatic residue at this position also for the interaction of the OCRE domain with SmB, as the Y495A mutation reduced the interaction between RBM5 OCRE and SmB while the Y495F retained binding competence (*Figure 2E*).

To probe the functional effects of the OCRE domain mutations, we tested the splicing activity of full-length RBM5 harboring mutations in the ligand binding region of the OCRE domain (*Figure 5*; *Figure 5—figure supplements 1* and *2*). As shown in *Figure 1E*, substitutions of conserved tyrosines that are involved in SmN interactions (Tyr495 and Tyr497) impair the activity of RBM5 in Fas alternative splicing regulation (*Figures 5A,C*; *Figure 5—figure supplement 2*). Further mutations of aromatic residues, including tyrosines 470, 479, 486 and 488 by alanine compromise the Fas exon 6 skipping activity of RBM5 to different extents, consistent with the reduced binding of these mutants to SmN-derived proline-rich ligands (see above) and with the location of these residues in the binding pocket for the SmN poly-proline motif (*Figures 4*, *5A–C*).

Substitution of the important Tyr495 residue by threonine or glutamate also compromised activity, while replacement by other aromatic residues (phenylalanine or tryptophan) retained full activity in splicing assays (*Figure 5A–C*; *Figure 5—figure supplement 2*), consistent with sustained SmN interaction (*Figure 2*; *Figure 2—figure supplement 1*). This argues that, in agreement with the structural analysis, the aromatic nature of the tyrosine is a key feature of the splicing regulatory properties of the OCRE domain. In contrast, mutations of residues, Tyr454, Asp458, Glu501 (*Figure 1E*) and Ser468 and Asn483 (*Figure 5—figure supplement 1*), which are remote from the SmN binding site, had only moderate effects on the activity of RBM5 on *FAS* splicing. All the mutant proteins accumulated to levels similar as the wild type (*Figure 5B*). These data further argue that specific recognition of the proline-rich SmN peptide by a cluster of aromatic residues in the RBM5 OCRE domain is required for the function of RBM5 as a splicing regulator.

We next wished to probe the contributions of different regions in the SmN/B/B' proteins for *FAS* alternative splicing. Consistent with previous results (*Saltzman et al., 2011*), we observed that the knock down of the SmN/B/B' proteins by siRNA in HeLa cells led to an increase in the level of *FAS* exon 6 skipping in endogenous transcripts or in transcripts derived from a Fas reporter (*Figure 5—figure supplement 3A,C*). These effects were attenuated by strengthening *FAS* exon 6-associated 5' splice site (*Figure 5—figure supplement 3C*), as has been previously reported (*Saltzman et al., 2011*). Co-expression of SmN with a *FAS* alternative splicing reporter bearing a mutation (Fas U-20C) that increases *FAS* exon 6 skipping (*Izquierdo et al., 2005*), led to increased levels of exon inclusion, while neither expression of the amino terminal part nor of the C-terminal part of the protein did (*Figure 5—figure supplement 3B*). While expression of SmN variants harboring mutations in the proline-rich stretches (either to glycine or to APAP motif) does not significantly compromise exon inclusion, mutation of the arginines flanking stretches of four prolines does (*Figure 5—figure supplement 3D*). However, the results of SmN/B/B' overexpression experiments are highly variable and difficult to interpret, perhaps because of additional complex effects of Sm protein overexpression on snRNP biogenesis/activity.

Taken together, the structural and functional analyses show that three tyrosine residues in the RBM5 OCRE domain (Tyr479, Tyr488 and Tyr495) play crucial roles in the recognition of proline-rich regions present in the conserved C-terminal tails of SmN/B/B' and reveal a tight correlation between this binding and the activity of RBM5 as a regulator of *FAS* alternative splicing.

We have previously shown that the alternative splicing activity RBM5 depends on the C-terminal region of RBM5 and that recombinant RBM5 inhibits the transition from the pre-spliceosomal complex A to spliceosomal complex B (*Bonnal et al., 2008*). To assess which region of the RBM5 protein is required, we performed *in vitro* spliceosome assembly assays with AdML pre-mRNA, using the N- and C-terminal halves of RBM5 as well as with a C-terminal version that lacks the OCRE domain or a fragment that includes only the OCRE and KEKE domains (*Figure 6—figure supplement 1*). These and additional transient transfection experiments (*Figure 6—figure supplement 2*) confirm that the inhibition of complex B formation depends on the C-terminal region of RBM5 and reveal that the OCRE domain is necessary but not sufficient for this activity. This is consistent with a key role for the

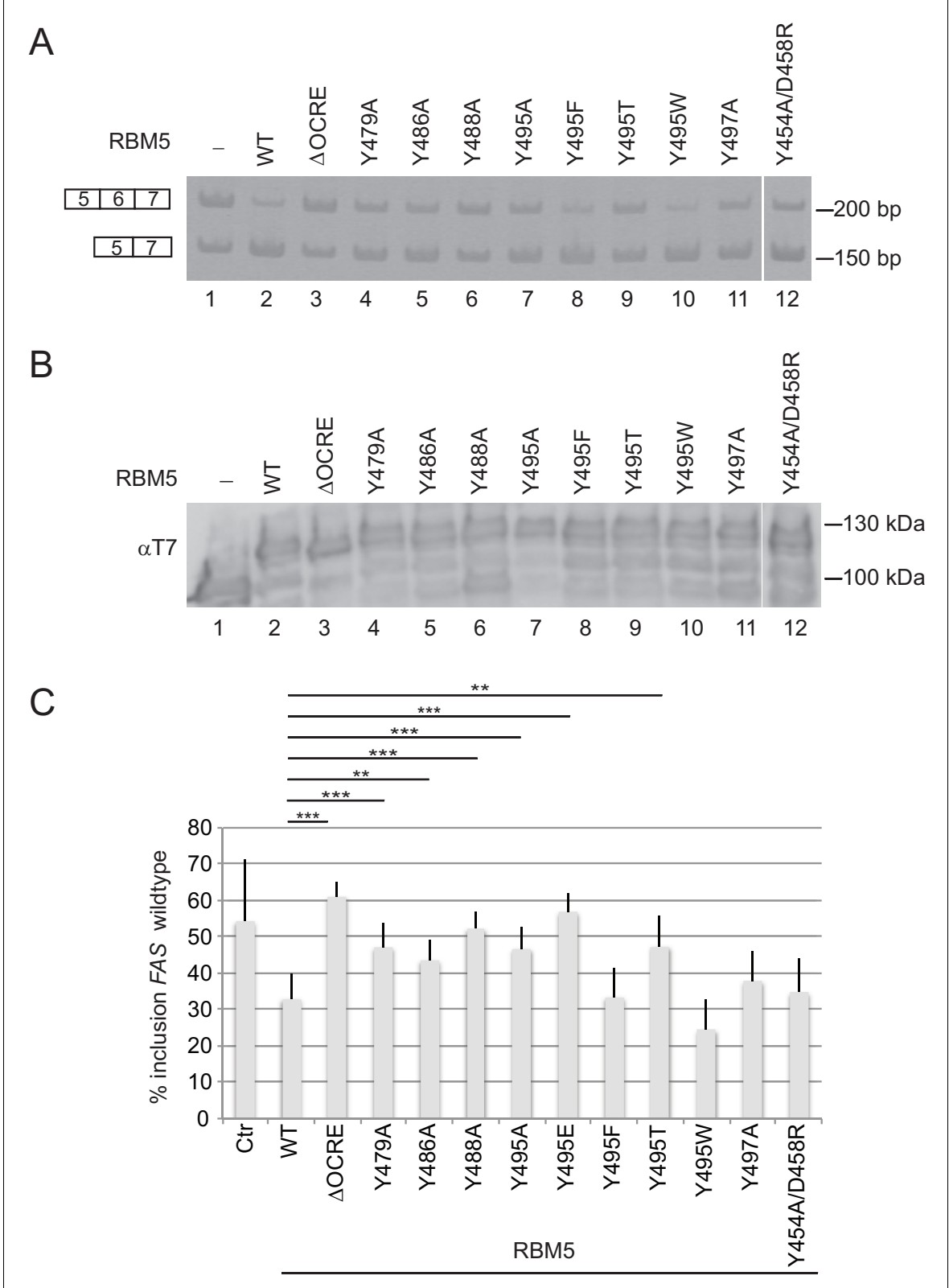

**Figure 5.** Mutational analysis of the RBM5 OCRE – SmN/B/B' interaction. Specific mutations of conserved residues in the RBM5 OCRE domain impair the activity of the protein in *FAS* alternative splicing regulation ex vivo. HeLa cells were co-transfected with a *FAS* alternative splicing reporter and T7-RBM5 expression plasmids (wild type or mutations of tyrosine residues as indicated). RNA and proteins were isolated 24 hr after transfection. Patterns of alternative splicing were studied by RT-PCR using specific primers. (**A**) Inclusion and skipping products are annotated. (**B**) Protein expression was

*Figure 5 continued on next page*

*Figure 5 continued*

detected by western blot with an anti-T7 epitope antibody. (**C**) Quantification of the activity of RBM5 OCRE domain mutants in *FAS* alternative splicing regulation of 3 to 10 replicates of the experiment. The percentage of inclusion is presented. T-test (two-tailed distribution, homoscedastic) results are mentioned (**<0,01; ***<0,001).

DOI: https://doi.org/10.7554/eLife.14707.016

The following figure supplements are available for figure 5:

**Figure supplement 1.** Effect of mutation of non-aromatic residues in RBM5 OCRE on FAS alternative splicing regulation.

DOI: https://doi.org/10.7554/eLife.14707.017

**Figure supplement 2.** Effects and statistical analysis of RBM5 mutations on FAS alternative splicing.

DOI: https://doi.org/10.7554/eLife.14707.018

**Figure supplement 3.** Effects of SmN wild type and mutants expression on *FAS* alternative splicing.

DOI: https://doi.org/10.7554/eLife.14707.019

interaction of OCRE with the PRM in the spliceosomal tails in the tri-snRNP, but argues that additional interactions involving the C-terminal region of RBM5 are important, including for example the previously reported interaction with U2AF65 (*Bonnal et al., 2008*).

## Discussion

The Octamer Repeat (OCRE) domain of RBM5 adopts a unique three-dimensional fold with a large number of conserved tyrosine residues. Although the primary sequence signature could suggest a linear sequence of octamer repeats, our structural analysis clearly reveals that the OCRE domain adopts a small globular fold. We show that the twisted β-sheet of the OCRE domain exposes conserved tyrosine residues that provide a platform for the recognition of a proline rich motif (PRM). This interaction is based upon a network of tyrosine residues as well as anchoring residues located in the loops of the β-strands. The SmN PRM is flanked by a positively charged arginine residue preceding the four consecutive prolines, followed by a hydrophobic residue and another arginine. Taking into account the sequence conservation, binding studies (*Figure 3*), and the details of molecular recognition of the SmN peptide (*Figure 4*), we propose that the OCRE domain recognizes a RPPP(P) GφR consensus motif. A key feature of the motif is a central PPII helix, which is recognized by a network of parallel and aromatic T-stacking of Tyr470, Tyr479, Tyr482, Tyr488 and Tyr495 with the proline side chains in the PPII helix. These tyrosines are exposed at one side of the OCRE β-sheet and collectively embed the PPII helix. Specific contacts that define the orientation of the bound peptide are mediated by two flanking arginine residues, with a stronger contribution of the N-terminal arginine (SmN Arg221, *Figure 3B*). The glycine residue may enable an unusual backbone conformation such that hydrophobic residues (φ, Ile/Met/Val flanking different PRMs in the SmN/B/B' C-terminal tails, *Figure 3A*) can shield the central hydrophobic PPII motif from solvent exposure. Notably, unbound PRMs harboring 3–4 consecutive prolines already adopt (at least partially) a preformed PPII conformation and thus reduce the entropy loss associated with binding to the OCRE domain. The somewhat reduced affinity for peptides with only three prolines may reflect a smaller propensity of forming PPII conformation in the free ligands and the lack of a preceding arginine, which is consistently present in all four proline PRMs.

The tertiary fold and the mode of PRM recognition by the OCRE domain is distinct from other PRM binding domains, such as GYF, SH3, and WW domains (*Figure 7A*). Although – as a common feature – aromatic residues are used to interact with proline residues and to recognize the PPII helix (*Ball et al., 2005*), different secondary structure elements are found in GYF (α-helical), SH3 (α/β fold), WW (β-sheet) and OCRE domains (β-sheet). Interestingly, a somewhat related PRM motif in the CD2 protein (PPPPGHR) was reported to interact with the GYF domain of the CD2BP2 protein (*Freund et al., 2002*). The CD2BP2 GYF domain and the FBP21 WW domain have been previously implicated in binding to the PRM sequences of SmB, suggesting that interactions between PRMs and these factors may be critical for the function of these proteins in spliceosome regulation (*Bedford et al., 1998*; *Klippel et al., 2011*).

An interesting aspect of the OCRE/Sm interaction is that the presence of multiple PRM motifs greatly enhances the overall affinity ranging from $K_D \approx$ mM for a single PRM peptide to $K_D = 20$–$40$ µM for complete Sm tails, and thus comparable to other PRM binding domains. The relatively weak

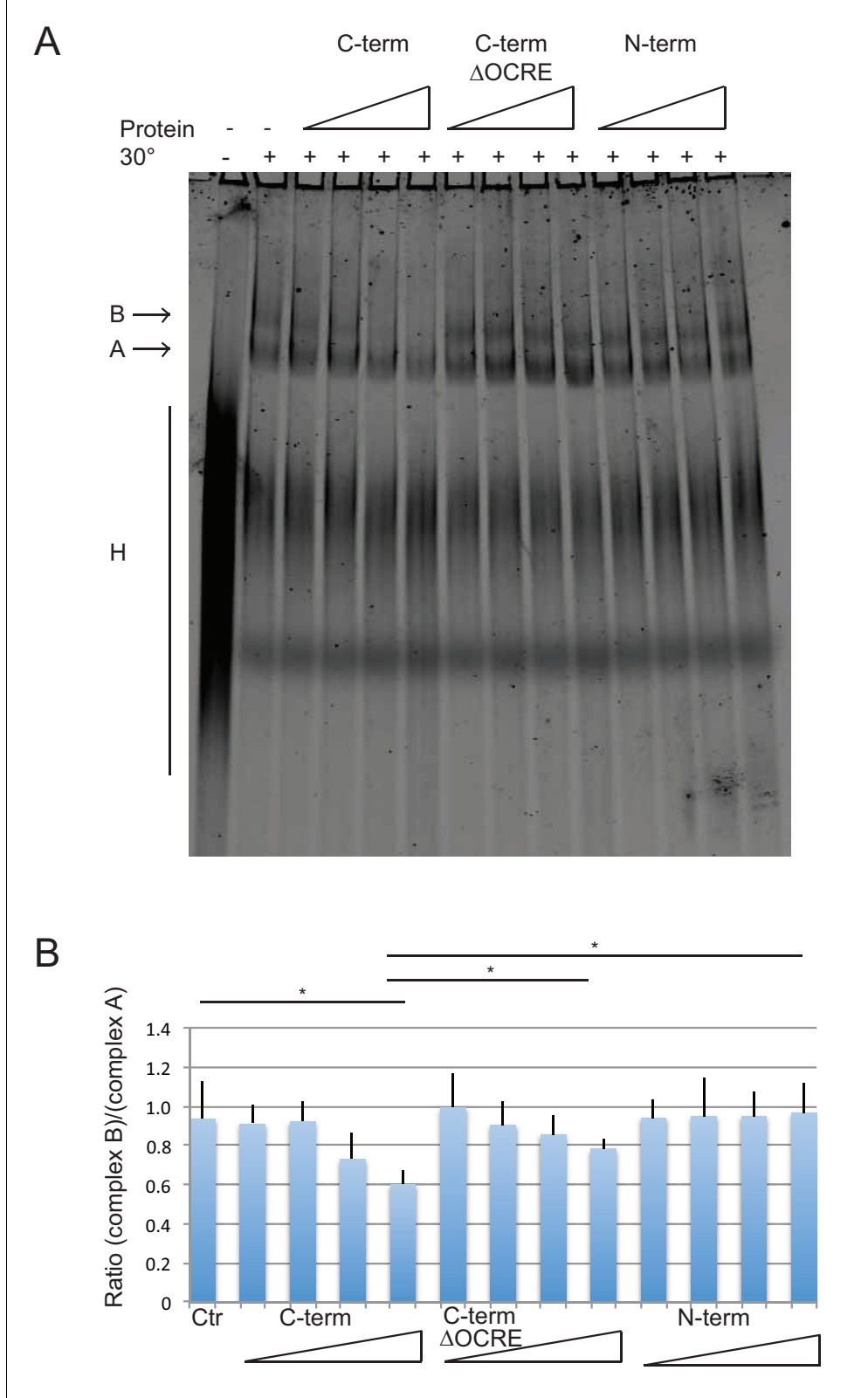

**Figure 6.** Spliceosome assembly assays with different regions of RBM5. (**A**) RBM5 inhibits the full spliceosome assembly on AdML and blocks the transition from complexes A to B in an OCRE dependent manner.Splicing complexes assembled on AdML were resolved by electrophoresis on native gel in the presence or absence of 25, 50, 100 and 200 ng/ul of the indicated proteins. The position of the H, A and B complexes are indicated. (**B**)
*Figure 6 continued on next page*

*Figure 6 continued*

Quantification of the activity of spliceosome assembly of the C-term, N-term and ΔOCRE proteins. The experiment was performed three times and the results of a T-test (two-tailed distribution, homoscedastic) are indicated (*<0,05).

DOI: https://doi.org/10.7554/eLife.14707.020

The following figure supplements are available for figure 6:

**Figure supplement 1.** Effects of RBM5 full-length and fragments on spliceosome assembly.

DOI: https://doi.org/10.7554/eLife.14707.021

**Figure supplement 2.** Effects of RBM5 full-length and OCRE on FAS alternative splicing.

DOI: https://doi.org/10.7554/eLife.14707.022

affinity of a single PRM found in the Sm tails argues that avidity effects due to the presence of multiple motifs play an important role for high affinity binding by increasing the local concentration of PRM motifs available for OCRE binding. We note that avidity effects provided by multiple binding sites have also been observed for other PRM binding domains (*Varani et al., 2000*; *Klippel et al., 2011*), and that this feature is reminiscent of the recognition of dimethyl-arginine residues in the Sm tails by Tudor domains (*Tripsianes et al., 2011*). On the other hand, the weak binding affinity of a single motif indicates that the interaction between RBM5 and SmN/B/B' may have been selected to be transient, as expected for a regulatory interaction that needs to promote interactions with the spliceosome but also needs to be disrupted at later steps of the splicing reaction.

Previously, we have shown that the splicing regulation by RBM5 acts at the stage of spliceosomal B complex formation (*Bonnal et al., 2008*). The role of RBM5 and its OCRE domain in this context could be to attract the U4/U6.U5 tri-snRNP to the splice sites. The fact that the SmN/B/B' tails are intrinsically disordered and extend very far away from the seven-membered Sm core ring in U snRNPs (*Figure 7B,C*), is consistent with such an activity. In fact, the recent EM structure of the tri-snRNP (*Nguyen et al., 2015*) revealed that the Sm cores of U4 and U5, as well as the LSm ring of U6 are located at the outside of this large RNP (*Figure 7C*). Thus, the C-terminal tails of SmN/B/B' of the U4 and U5 components are clearly accessible and could scan for possible binding partners of their proline-rich motifs, such as the RBM5 OCRE domain. This could help to localize the tri-snRNP to the splice sites that are regulated by RBM5.

SmB was described to participate in alternative splicing of its own pre-mRNA and many additional genes, including the *FAS* gene (*Saltzman et al., 2011*). Recently, mutations in SmB were linked to cerebro-costo-mandibular syndrome (*Bacrot et al., 2015*). The multi-domain protein RBM5 promotes *FAS* exon 6 skipping and it has been proposed that for this activity the protein modulates splice site pairing after the competing 5' and 3' splice sites have been recognized by U1 and U2 snRNPs, respectively (*Bonnal et al., 2008*). This model is based upon the observation that RBM5 inhibits the transition from A (U2 snRNP binding) to B (U4/U6.U5 tri-snRNP binding) complex formation in the introns flanking exon 6. It is conceivable that interactions mediated by the C-terminal tails of SmN/B/B', which are present in U1, U2 and the tri-snRNP, facilitate the transition from A to B complex and that binding of the OCRE domain of RBM5 prevents these interactions and therefore the progression of spliceosome assembly. Direct or indirect association of RBM5 with particular regions of the pre-mRNA – likely involving other regions of the protein including RRM or zinc finger domains – may prevent A to B transition of pre-spliceosomes assembled on the flanking introns of exon 6, but fail to prevent tri-snRNP assembly on the distal splice sites, thus facilitating exon skipping. Alternatively, or in addition, RBM5 interaction with the C-terminal tails of SmN/B/B' may lead to a general decrease in snRNP function, possibly more acute for U2 snRNP. As decreased activity of U2 snRNP – e.g. by individual knock down of its protein components – is known to result in increased *FAS* exon skipping (*Papasaikas et al., 2015*), RBM5-mediated reduction in U2 function could potentially explain at least part of the effects of the protein on *FAS* splicing regulation.

Taken together, our study provides a molecular understanding of how the OCRE domain of RBM5 interacts with proline-rich sequences of the SmN/B/B' tail and thus identifies a key interaction essential for regulation of alternative splicing.

It is interesting to note that, based on primary sequence alignments, the OCRE domains of RBM5 and RBM10 are expected to adopt highly similar structures and possibly functions. In fact, the recently reported structure of the RBM10 OCRE domain is highly similar to RBM5 OCRE (backbone coordinate r.m.s.d. 1.1Å) (*Martin et al., 2016*). Interestingly, mutation of a conserved tyrosine

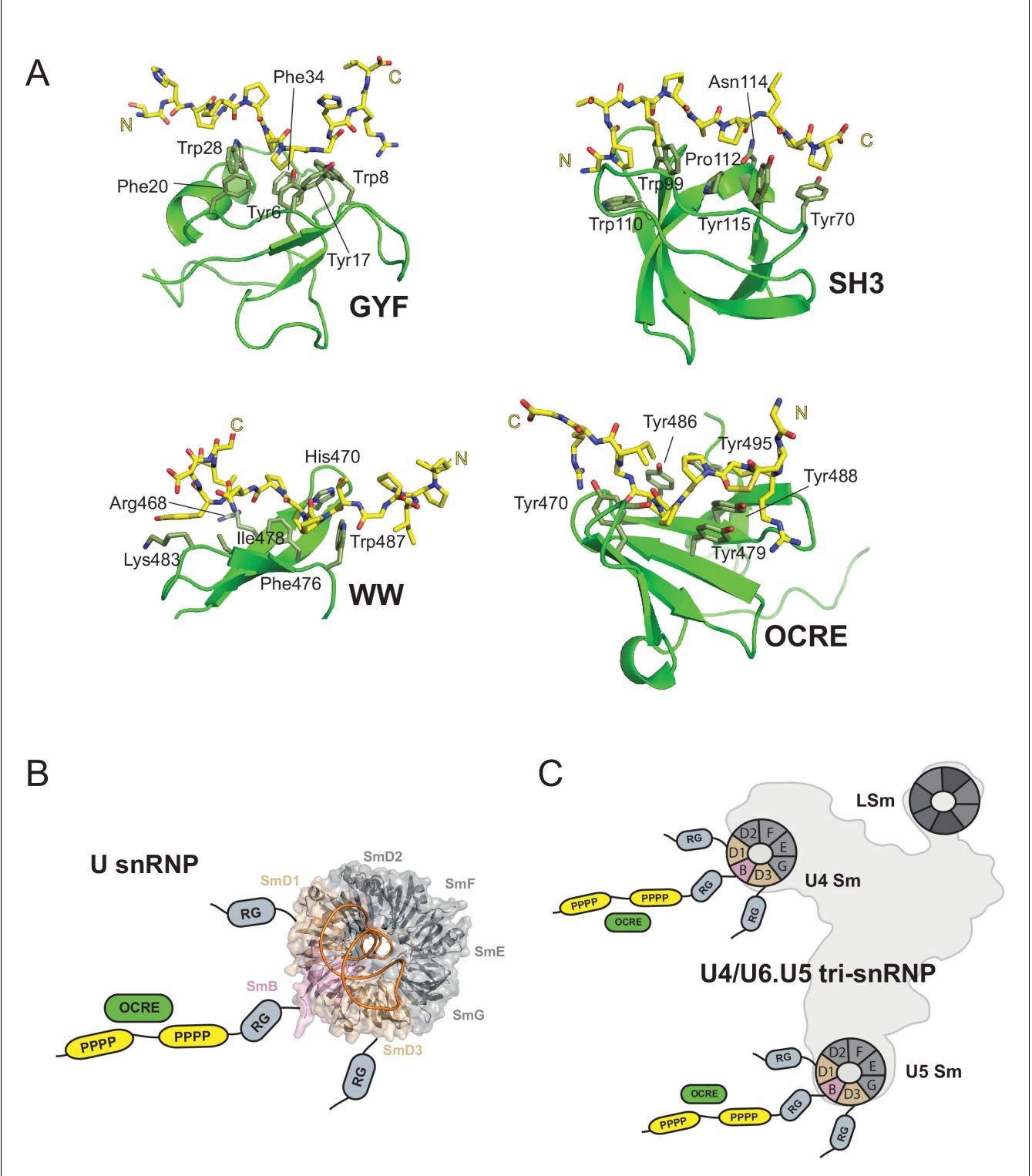

**Figure 7.** OCRE is a novel PRM-binding domain that interacts with snRNPs. (**A**) Comparison of proline-rich motif (PRM) recognition by PRM binding domains. Cartoon representations of various PRM binding domains (green) with side chains of important residues shown as sticks and annotated. PRM ligands are shown by stick representation (yellow). PDB accession codes: GYF (PDB 1L2Z), SH3 (PDB accession 1ABO), WW (PDB 1I5H), and RBM5 OCRE. (**B**) The C-terminal tail of SmN/B/B' comprising RG-rich regions and proline-rich motifs (PRM) extends far from the core Sm core fold in the seven-membered Sm ring of assembled U snRNPs. The crystal structure of U4 snRNP core (PDB 4WZJ) (*Leung et al., 2011*) is shown, the C-terminal tails of SmD1/D3, which harbour RG-rich regions and SmN/B/B' comprising RG and PRM regions are not visible in the crystal structure and indicated

*Figure 7 continued on next page*

*Figure 7 continued*

schematically. (C) The seven-membered Sm core rings of U4 and U5 snRNPs are located at the outside of the assembled U4/U6.U5 tri-snRNP (*Nguyen et al., 2015*). Thus, also in the tri snRNP the SmN/B/B' tails are accessible for interactions with the RBM5 OCRE domain.

DOI: https://doi.org/10.7554/eLife.14707.023

residue (corresponding to Tyr470 in RBM5, where it contributes to the recognition of the PRMs by the OCRE domain) is frequently found associated with lung carcinoma (*Imielinski et al., 2012*). It is thus tempting to speculate that splicing defects that are linked to impaired interactions with PRM in SmN/B/B' proteins contribute to the pathogenesis.

## Materials and methods

### Cloning, protein expression and purification

RBM5 (Swiss Prot P42756) OCRE domain (451–511) and all the mutants were subcloned into a pETM11 vector (with a N-terminal $His_6$-tag) from the corresponding GST-RBM5 full-length DNA. The several SmN and SmB constructs were cloned into a pETM30 vector, containing a N-terminal $His_6$-tag followed by glutathione S-transferase protein. In both vectors, a TEV site is present before the corresponding protein. All the proteins were produced by overexpression in *E. coli* BL21 cells at 20°C for 16 hr after induction with 0.5 mM of IPTG when cells were at approximately O.D. 0.7 in media supplemented with 30 μg/ml kanamycin. For unlabeled proteins, bacteria were grown in Luria broth.

For isotope-labeled proteins, bacteria were grown in M9 minimal media supplemented with $^{13}C$-glucose and/or $^{15}NH_4Cl$. Cell lysates were suspended in buffer containing 20 mM Tris pH 8, 300 mM NaCl, 5% glycerol, 10 mM imidazole, and 2 mM of 2-mercaptoethanol and purified with Ni-NTA Superflow beads (Qiagen, Hilden, Germany) using standard conditions. After overnight cleavage of the fusion protein with tobacco etch virus protease, proteins were purified in a gel fil-tration column 26/60 sephadex II (GE Healthcare, München, Germany) and buffer was exchanged to NMR buffer; 20 mM sodium phosphate pH 6.5, 50 mM NaCl. For measurements in $D_2O$, the protein was lyophilized and dissolved in $D_2O$. The OCRE domain used for NMR studies thus com-prises residues 451–511 preceded by a GAM tripeptide that results from the TEV cleavage. The SmN 219–229 peptide (Peptide Specialty Laboratory, Heidelberg, Germany) was dialyzed against water, lyophilized and then dissolved in NMR buffer.

GST-tagged RBM5 OCRE domain proteins were expressed in BL21 cells. Bacteria were trans-formed and grown in one liter LB to an absorbance at 600 nm of 0.6 before the induction of the expression with 1 mM IPTG for 3 hr at 37°C. The pellet was resuspended in cold buffer X (20 mM Tris pH 7.5; 1 M NaCl; 0.2 mM EDTA; 1 mM DDT and protease inhibitors cocktail (Roche Diagnos-tics, reference 11697498001, Mannheim, Germany) and sonicated. The supernatant was collected after centrifugation for 20 min at 10,000 rpm at 4°C and incubated with Glutathione Sepharose 4B beads for 15 min on a rotating wheel at 4°C. The beads were washed with 50 ml buffer X 3 times 10 min and eluted on column with 50 mM glutathione, 100 mM Hepes pH 8,0; 1 mM DTT. The selected fractions were dialyzed against buffer D (20 mM Hepes pH 8.0; 20% glycerol; 0.2 mM EDTA; 0.1 M KCl; 1 mM DTT and 0.01% NP40), frozen in liquid nitrogen and stored at −80°C. T7-tagged SmN protein was expressed and purified from HEK 293T cells (*Cazalla et al., 2005*).

The N-terminal (aa 1–318) and C-terminal (320–815) regions of RBM5 were produced by PCR amplification using specific primers and cloned into the pET15b vector (Novagen) generating plas-mids N-term and C-term, respectively. The plasmid C-term ΔOCRE was obtained by removing the aa 452 to 511 of plasmid C-term using site-directed mutagenesis. Recombinant $His_6$-tagged proteins were expressed in *E. coli* (BL21-CodonPlus), solubilized under denaturing conditions (6M Guanidine hydrochloride) (*Wingfield et al., 2001*) and purified by Ni-NTA affinity chromatography. After purifi-cation, the proteins were dialysed overnight against buffer D using Slide-A-Lyser devices (Pierce).

### NMR spectroscopy

Spectra were recorded at 298 K in DRX500, 600 and 900 spectrometers with cryogenic triple reso-nance probes using $^{13}C$, $^{15}N$ labeled OCRE sample (1 mM) for the apo structure, and a $^{13}C$, $^{15}N$

labeled OCRE and unlabeled peptide (2 mM and 14 mM, respectively) for the complex. Data were processed with NMRPipe (*Delaglio et al., 1995*) and analyzed using NMRView (*Johnson and Blevins, 1994*). For backbone resonance assignments, standard experiments were recorded, including 3D HNCA, HNCACB, HN(CO)CACB experiments. Side chain resonances assignments were made with 3D HCCH-TOCSY and 3D 15N-HSQC TOCSY experiments. Distance restraints were derived from $^{15}$N- and $^{13}$C-resolved three-dimensional, $^{1}$H homonuclear two-dimensional (mixing time 120 ms). In addition, for the complex structure, assignment and distance restraints information for the peptide were obtained by recording a $^{15}$N, $^{13}$C filtered TOCSY with the mixing times of 15, 30, 60, 90 ms and NOESY experiments with NOE mixing times of 120 ms. H/D exchange experiments were recorded after lyophilization. $^{15}$N relaxation experiments ($T_1$, $T_2$ and {$^{1}$H}–$^{15}$N heteronuclear NOEs) were measured for both the apo and bound form of OCRE. Chemical shift mapping on OCRE was done by monitoring the 2D $^{1}$H,$^{15}$N HSQC, 2D $^{15}$N-labeled OCRE (0.2 mM) with an excess of unlabeled SmN constructs until no further changes in chemical shifts were observed in the 2D $^{1}$H,$^{15}$N HSQC spectra. Combined CSPs were calculated as $\Delta\delta$ (ppm) = $((10{*}\Delta\delta_{HN})^2 + (\Delta\delta_N)^2)^{1/2}$.

## Determination of relative binding affinity for peptides from NMR titrations

A semi-quantitative approach was taken to assess the contribution of the amino acid sequence of PRM motifs in Sm tails to the binding interaction. Due to the relatively weak interaction, NMR titrations were used to compare relative affinities between wild type peptides and peptides with specific mutations. The SmN derived wild type and mutant peptides were titrated into OCRE at a 1:10 ratio of OCRE:peptide. CSPs from 7 OCRE residues (Y470, Y471, Y479, D481, N483, S490, Y495) were added for each of the peptide titration and normalized with that of the wild type peptide (GMRPPPPGIRG). The score is used to compare the relative affinities of the various peptides with the wild type (*Figure 3B*). An overlay of $^{1}$H,$^{15}$N-HSQC spectra of OCRE bound to wild type (GMRPPPPGIRG) or variant peptides -GM**A**PPPPGIRG, GM**E**PPPPGI**E**G illustrates the different extent of CSPs observed for seven residues in the OCRE domain that are most strongly affected by the binding (*Figure 3—figure supplement 2*). These residues were selected to define the CSP score.

## Structure calculations

For the apo structures, automatic NOE assignments and structure calculations were initially performed by CYANA3 (*Güntert, 2009*). Subsequently, NOEs were manually checked and applied as distance restraints together with dihedral angle restraints in a simulated annealing protocol using ARIA (*Linge et al., 2001*) and CNS (*Brunger et al., 1998*). Dihedral restraints were derived from TALOS+ (*Shen et al., 2009*), hydrogen bond distance restraints were applied based on secondary structure identified by TALOS+, and added during structure calculations. For the complex structure, manual assigned NOEs were applied as distance restraints together with dihedral angle and measure hydrogen bond restraints in a simulated annealing protocol using ARIA (*Linge et al., 2001*) and CNS (*Brunger et al., 1998*). Water refinement was performed on the final ensembles of NMR structures (*Linge et al., 2003*). The structural quality of the 10 lowest energy structures out of 100 calculated structures was evaluated using ProcheckNMR (*Laskowski et al., 1996*), and the iCING (*Doreleijers et al., 2012*) and PSVS (*Bhattacharya et al., 2007*) servers. Ramachandran statistics for the free RBM5 OCRE domain and the complex structure with SmN (219–229) are 91.5%/8.5%/0%/0% and 88.0%/9.8%/2.0%/0.2% in the most favored/additionally/generously/disallowed regions, respectively. Ribbon representations and the electrostatic surface potential were prepared with PYMOL (DeLano Scientific, San Carlos, CA, USA).

## Isothermal titration calorimetry

ITC experiments were performed using an ITC200 instrument (MicroCal, Wolverton Mill, UK) at 24°C. SmN constructs (168–196, 168–240) and an SmN peptide (219–229) at 1 mM, 1 mM and 5 mM, respectively, were titrated into OCRE (100 µM, 100 µM and 1 mM, respectively). For the mutation analysis, purified OCRE mutants (2 mM) were titrated to wild type SmN (29.5 µM) or SmB (40 mM). All purified proteins were in the same buffer, 20 mM sodium phosphate pH 6.5, 50 mM NaCl. The lyophilized peptide was dialyzed against water, lyophilized again, and then dissolved in

the same buffer as the protein. The heat of dilution was measured by titrating SmN or OCRE mutants into buffer. The titration protocol consisted of one initial injection of 0.4 µL followed by 38 injections of 1 µl of the ligand into the protein sample with intervals of 120 s, allowing the titration peak to reach the baseline. Data were calculated using the program Origin v7.0 (MicroCal) and duplicates were measured for all the experiments.

## Circular Dichroism (CD) spectroscopy

All CD spectra were recorded on a JASCO-J715 spectropolarimeter and analyzed with Spectramanager version 1.53.00 (Jasco Corp.). The temperature was regulated using a Peltier type control system (PTC-348WI). The spectra were recorded at 5°C in 20 mM sodium phosphate,100 mM NaCl, pH 6.5 buffer from 190–260 nm wavelength with a 1.0 nm bandwidth, 0.5 nm pitch at a scan speed of 50 nm/min, in cuvettes with 0.1 cm path length. All spectra are presented as an average of 20 scans, obtained after buffer subtraction and plotted as mean residue ellipticity (deg $cm^2$ $dmol^{-1}$) vs wavelength (nm). All peptides were measured at 0.3 mM concentration in a buffer containing 20 mM sodium phosphate, 100 mM NaCl, pH 6.5 at 5°C.

## Glutathione S-transferase pull-down experiments

GST RBM5 OCRE and T7 SmN or [35]S-labeled SmB proteins were incubated for one hour at 4°C degrees on a rotating wheel in 1 ml PBS supplemented with 0.1% Triton X100. 45 µl of packed and equilibrated GSH beads (Glutathione Sepharose 4B, GE Healthcare, reference 17-0756-05) were added and the samples were incubated for one hour more as before. The beads were then washed four times with 1 ml PBS-0.1% Triton X100 and the proteins were directly eluted in SDS loading dye at 95°C for 5 min under shaking, loaded on SDS gels, separated by electrophoresis. Proteins were revealed by autoradiography or analyzed by western blot using the following antibodies: anti-T7 (T7-Tag Antibody HRP conjugate, Novagen, reference 69048) and anti-GST (GST(B14)-HRP mouse monoclonal, Santa Cruz Biotechnology, reference sc-138 HRP).

## Ex vivo splicing analysis

Protocol of co-transfection, RT-PCR and western blot analysis were carried out as previously described (*Bonnal et al., 2008*). siRNA against Sm proteins were carried out as described (*Saltzman et al., 2011*).

## Spliceosome assembly assays

Cy5-CTP labeled AdML RNA bearing exon 1 – intron 1 – exon 2 was *in vitro* transcribed using T7 Megascript kit (Ambion). The spliceosome assembly reaction was performed as described previously (*Mackereth et al., 2011*) with 10 ng/ul fluorescently labeled RNA and the indicated recombinant protein. After electrophoresis, the gel was analyzed directly with a PhosphorImager Typhoon. Quantification was carried out using Image Quant.

## Yeast two hybrid assays

The C-terminal domain of human SmB (aminoacids 84–231; Accession n°: NM_003091) was PCR-amplified using oligonucleotides carrying EcoRI (Forward) and SalI (Reverse) restriction sites and the pGFP-huSmB plasmid as template (*Mouaikel et al., 2003*). After agarose gel electrophoresis, the DNA fragment was purified using the GeneClean procedure and transferred into EcoR1-Sal1 cut pGBT9 vector carrying the DNA binding domain of Gal4 (*Fields and Song, 1989*).

The various pACT2-RBM5 plasmids containing the RBM5 full length and mutated coding sequences in frame with the Gal4AD (activation domain) were constructed using the Gateway system and a pACT2-based vector according to the manufacturer's instructions (Invitrogen). The plasmids used for amplification have been described previously (*Bonnal et al., 2008*). All pDonor constructs were sequenced prior to proceed to LR recombination. The sequences of cloning junctions and coding sequences of all plasmids were verified to ensure the absence of any unwanted mutations.

Yeast strains were grown using standard procedures and media. For 2YH assays, the pGBT-CterSmB plasmid and the appropriate pACT2 plasmids were transformed into the CG1945 strain (*Fromont-Racine et al., 1997*). Transformants were selected on double selectable media (-Leu -Trp) and further grown in minimal -Trp -Leu liquid medium. Growth of yeast was measured by spotting

serial dilutions of liquid cultures on -Leu -Trp -His plates which enables selection of interacting partners.

## Accession codes

The atomic coordinates for the NMR ensembles of the RBM5 OCRE domain and the complex with the SmN (219–229) are deposited in the Protein Data Bank under accession numbers 5MFY and 5MF9, respectively. The chemical shift assignments have been deposited in the Biological Magnetic Resonance Data Bank under accession numbers 34068 and 34067.

## Acknowledgements

The authors are grateful to Reinhard Lührmann for discussions. We thank the Bavarian NMR Centre (BNMRZ) for NMR measurement time. LW acknowledges support by an EMBO Long Term Fellowship ALTF 1520-2011 and NIH Grant P20GM103408; KS for support by the IMPRS-LS Graduate School. This work was supported by the *Deutsche Forschungsgemeinschaft* (DFG SFB1035, GRK1721) to MS, and by the *Centre National de la Recherche Scientifique* (CNRS) to RB. Work in JV's lab is supported by *Fundación Botín*, by *Banco de Santander* through its Santander Universities Global Division and *Consolider RNAREG*, *MICINN* and *AGAUR*, and by the *Spanish Ministry of Economy and Competitiveness* 'Centro de Excelencia Severo Ochoa 2013-2017', SEV-2012-0208. Work in JV's lab was also supported by the European Research Council (ERC AdG - GA670146 - MASCP).

## Additional information

### Competing interests

Juan Valcárcel: Reviewing editor, *eLife*. The other authors declare that no competing interests exist.

### Funding

| Funder | Grant reference number | Author |
| --- | --- | --- |
| Centre National de la Recherche Scientifique | | Rémy Bordonné |
| Fundación Botín | | Juan Valcárcel |
| Ministerio de Economía y Competitividad | Centro de Excelencia Severo Ochoa 2013-2017, SEV-2012-0208 | Juan Valcárcel |
| Banco de Santander | Consolider RNAREG, MICINN, AGAUR | Juan Valcárcel |
| European Research Council | ERC AdG - GA670146 - MASCP | Juan Valcárcel |
| Deutsche Forschungsgemeinschaft | SFB1035 | Michael Sattler |
| Deutsche Forschungsgemeinschaft | GRK1721 | Michael Sattler |

The funders had no role in study design, data collection and interpretation, or the decision to submit the work for publication.

### Author contributions

André Mourão, Analysis and interpretation of data, Drafting or revising the article, Conception and design, Acquisition of data; Sophie Bonnal, Lisa Warner, Conception and design, Acquisition of data, Analysis and interpretation of data, Drafting or revising the article; Komal Soni, Acquisition of data, Analysis and interpretation of data; Rémy Bordonné, Conception and design, Acquisition of data, Analysis and interpretation of data; Juan Valcárcel, Michael Sattler, Conception and design, Analysis and interpretation of data, Drafting or revising the article

**Author ORCIDs**

Michael Sattler http://orcid.org/0000-0002-1594-0527

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
