## [Decision Letter]

Thank you for submitting your article "Structural basis for the recognition of spliceosomal SmN/B/B' proteins by the RBM5 OCRE domain in splicing regulation" for consideration by *eLife*. Your article has been favorably evaluated by James Manley as the Senior Editor and three reviewers, one of whom, Douglas Black, is a member of our Board of Reviewing Editors.

The reviewers have discussed the reviews with one another and the Reviewing Editor has drafted this decision to help you prepare a revised submission.

Summary:

This paper from the Sattler and Valcárcel groups examines the interaction of splicing regulatory proteins of the Rbm5 family with the SmN/B/B' proteins within the core of the Sm snRNPs. There are many regulatory proteins that bind to the pre-mRNA to alter spliceosome assembly, but remarkably few interactions have been described between these regulators and core components of the spliceosome. Such information is key to understanding the mechanisms of splicing regulation. In previous work, the Valcárcel lab has characterized the Rbm5 family of splicing regulators, showing that they control important splicing events for apoptosis and cancer progression. They also reported that the octamer repeat (OCRE) domain of the Rbm5 protein interacts with the tri-snRNP complex that forms part of the mature spliceosome. OCRE domains containing repeats of aromatic residues are found in multiple proteins but their structure and interactions are unknown. Here the authors solve the structure of the Rbm5 OCRE by NMR, showing that it adopts an antiparallel β sheet structure with each repeat comprising a β strand. The conserved tyrosines project from both sides of the sheet. An N-terminal extension interacts with the residues on one face of the sheet to stabilize the domain, while tyrosines on the opposite face are available for other interactions. The authors define repeated proline-rich motifs within the C-terminal domains of the SmN/B/B' proteins as ligands for the OCRE domain. They show this interaction requires particular aromatic residues on the interaction surface of OCRE. They define a consensus proline rich motif (PRM) on the SmN protein that is required for OCRE interaction, and which includes 3 or 4 continuous proline residues flanked by arginines. Solving the structure of the OCRE bound to this motif, they show that the PRM forms a proline type II helix and the OCRE engages the PRM through a combination of aromatic stacking interactions between the prolines and the tyrosines, and both hydrogen bonding and electrostatic interactions with the flanking arginines. Through coexpression of mutant Rbm5 with a reporter minigene they show that these OCRE residues are indeed needed for Rbm5 regulatory activity. Finally, they compare OCRE to other domains such as SH3 and WW that also use aromatic residues to recognize poly-proline motifs.

The reviewers agreed that this is a well-executed study that uses a range of approaches to illuminate new aspects of splicing, including the structure and interactions of OCRE domain and the role of the OCRE/SmB interaction in splicing regulation. A number of issues were raised that need to be addressed before the paper is suitable for publication.

Essential revisions:

1) It is shown that mutation of the 4 prolines of the polyproline motif to alanines does not impact binding affinity. This is attributed to the alanines' ability to adopt a polyproline type II helical conformation. However, the hydrophobic interactions described should still be weakened upon mutation of prolines to alanines. The authors should test substitutions to other hydrophobic amino acids, such as valine, to prove that the backbone conformation is the major determinant of binding affinity.

2) The results in Figure 3B are surprising. It is well accepted that the extent of chemical shift perturbation does not necessarily correlate with *K*_D_ in mutant analysis, as the mutant ligand can bind in a slightly different orientation. In addition, mutations have a large influence on the extent of the chemical shift perturbations observed for residues in direct contact. Figure 3B needs to be reinterpreted in light of this, and the authors need to demonstrate that much more clearly that their *K*_D_ changes are as they describe.

3) The model for how the Rbm5/SmB interaction is altering spliceosome assembly is not at all clear. They propose based on earlier work that Rbm5 is interacting with SmB within the tri-snRNP. Isn't it also possible that the interaction could occur between Rbm5 and SmB in the U1 and U2 snRNP's? In Bonnal et al., 2008, they show that GST-OCRE interacts with the 220 and 200 kDa proteins of the U5 snRNP when incubated in HeLa nuclear extract. However, the data in that paper is not sufficient to propose that it must be a tri-snRNP interaction that is relevant here. The most prominent protein identified in that study was the U1-70K protein, suggesting that the OCRE domain also binds to the U1 snRNP. OCRE could thus indirectly affect the recruitment of the tri-snRNP via the U1 interaction. Additional bands in the OCRE-domain pulldown were also not identified, and no information was provided about which snRNAs were pulled down. Thus, it is not clear if the OCRE domain can bind all or only some snRNPs containing SmB/B'/N. These questions are not mentioned in the present Discussion but dramatically affect any model for how OCRE alters spliceosome assembly. Given the nicely shown interaction of RBM5's OCRE domain with the proline-rich regions of SmB/B'/N, and the fact that these proteins are present in the U1, U2, U4 and U5 snRNPs, a more thorough analysis of the interaction partners of the OCRE domain in nuclear extract is needed. The pulldown analysis should be repeated, analyzing all proteins and RNAs that are bound by OCRE, as well as analyzing how the stringency of the buffers used during pulldown and wash affect the spectrum of interacting species.

4) Also in the previous study (Bonnal et al., 2008), addition of RBM5 to an in vitro splicing reaction led to the accumulation of A complexes, and a block to B complex formation on intron 5 or 6 of Fas pre-mRNA. This indicated that stable recruitment of the tri-snRNP was hindered in the presence of RBM5. What effect does a RBM5 OCRE domain alone, or an RBM5 mutant lacking OCRE or with an inactive OCRE domain have on splicing complex formation in vitro? This experiment would provide additional information about the mechanism whereby RBM5's OCRE domain affects splicing. For example, Rbm5 also interacts with U2AF. It is possible that this U2AF interaction is sufficient to block to tri-snRNP recruitment, and the OCRE/SmB interaction is serving other roles.

---

## [Author Response]

*[…] The reviewers agreed that this is a well-executed study that uses a range of approaches to illuminate new aspects of splicing, including the structure and interactions of OCRE domain and the role of the OCRE/SmB interaction in splicing regulation. A number of issues were raised that need to be addressed before the paper is suitable for publication.*

*Essential revisions:*

*1) It is shown that mutation of the 4 prolines of the polyproline motif to alanines does not impact binding affinity. This is attributed to the alanines' ability to adopt a polyproline type II helical conformation. However, the hydrophobic interactions described should still be weakened upon mutation of prolines to alanines. The authors should test substitutions to other hydrophobic amino acids, such as valine, to prove that the backbone conformation is the major determinant of binding affinity.*

Thank you for this suggestion. We have performed further experiments to test the contribution and properties of the proline side chains by substituting proline by valine and comparing binding affinities *in vitro*. Unfortunately, a 4V version of the peptide (GMRVVVVGIRG) was completely insoluble, and thus no data could be obtained for this variant. Likewise, the SmN protein harboring 4P/3P→4V/3V mutations could not be expressed and purified. Presumably, the bulky and hydrophobic valine side chains cause aggregation of the protein, as observed for the peptide *in vitro*.

As a perhaps less optimal, but feasible alternative to evaluate the importance of hydrophobic interactions involving the proline side chains for OCRE binding, we tested peptides and SmN variants where prolines were replaced by glycine. CD spectra of the 4G peptide (revised Figure 3—figure supplement 1) indicate that it adopts (at least partially) a PPII conformation, consistent with previous reports^1,2^.

However, the affinity of the 4P/3P→4G/3G variant SmN protein to the OCRE domain is significantly reduced (*K*_D_ = 192 ± 66 µM) compared to the wild type SmN protein (*K*_D_ = 41 ± 2 µM) (revised Table 2). This is consistent with a reduced PPII propensity but also a contribution of hydrophobic interactions involving proline side chains in the wild type SmN, which are not possible with the glycine mutants.

Consistent with the reduced affinity for the OCRE domain observed *in vitro*, the activity of the 4P/3P→4G/3G SmN protein to promote *FAS* alternative splicing is significantly impaired (revised Figure 5—figure supplement 3D), comparable to the reduced activity of the APAP mutant.

The results of the 4P/3P→4G/3G mutant are therefore consistent with the concept that adoption of a polyproline type II helical conformation by the C-terminal tail of SmN, and establishment of a defined set of hydrophobic interactions between proline residues and aromatic side chains in the OCRE fold are necessary for OCRE binding and for alternative splicing regulation.

References

1) Brown AM, Zondlo NJ. 2012. A propensity scale for type II polyproline helices (PPII): aromatic amino acids in proline-rich sequences strongly disfavor PPII due to proline-aromatic interactions. Biochemistry 51 (25):5041-5051. doi:10.1021/bi3002924

2) Kelly MA, Chellgren BW, Rucker AL, Troutman JM, Fried MG, Miller AF, Creamer TP. 2001. Host-guest study of left-handed polyproline II helix formation. Biochemistry 40 (48):14376-14383

*2) The results in Figure 3B are surprising. It is well accepted that the extent of chemical shift perturbation does not necessarily correlate with K_D_ in mutant analysis, as the mutant ligand can bind in a slightly different orientation. In addition, mutations have a large influence on the extent of the chemical shift perturbations observed for residues in direct contact. Figure 3B needs to be reinterpreted in light of this, and the authors need to demonstrate that much more clearly that their K_D_ changes are as they describe.*

We agree with the comment that an interpretation of chemical shift perturbations (CSP) in terms of affinity is approximate and has to be done with caution. We have resorted to this approach as no other experimental technique is available to provide an – at least semi-quantitative – measure of relative affinities. Given that the peptide ligands do not exhibit aromatic side chains themselves and considering that we know the structural details of the interaction, we believe that changes for the amide signals shown in Figure 3B report on the local changes in chemical shift and depend on the population of bound state, i.e. relative affinities.

Note, that the signals shown in Figure 3B were carefully chosen as they report on key interactions with the (wildtype) ligand. However, the affinity score was based on analyzing all amide signals that experience significant chemical shift changes (i.e. above a certain cut-off), thereby providing some averaging of local effects, which thus should correlate in a reasonable way with overall binding affinity. To illustrate the changes in CSPs for different amide signals, we show the NMR spectra for representative peptide ligands in new Figure 3—figure supplement 2. The figure also shows that the CSPs for the different peptides follow a linear vector, which suggests that only a 2-site process is sensed, i.e. free vs. bound states, as otherwise deviations from linearity of the CSP vectors would indicate more local variations depending on the residues analyzed.

We have carefully updated the description of the approach and added a cautionary statement about its interpretation. Please note that the same mutations were introduced into the complete tail peptide. The very good agreement between relative affinities determined for the complete tails by ITC and those determined for the peptides by our NMR analysis, suggests that the NMR data provide useful information, which was then useful to guide our mutational analysis of the recognition of a single peptide motif.

*3) The model for how the Rbm5/SmB interaction is altering spliceosome assembly is not at all clear. They propose based on earlier work that Rbm5 is interacting with SmB within the tri-snRNP. Isn't it also possible that the interaction could occur between Rbm5 and SmB in the U1 and U2 snRNP's? In Bonnal et al., 2008, they show that GST-OCRE interacts with the 220 and 200 kDa proteins of the U5 snRNP when incubated in HeLa nuclear extract. However, the data in that paper is not sufficient to propose that it must be a tri-snRNP interaction that is relevant here. The most prominent protein identified in that study was the U1-70K protein, suggesting that the OCRE domain also binds to the U1 snRNP. OCRE could thus indirectly affect the recruitment of the tri-snRNP via the U1 interaction. Additional bands in the OCRE-domain pulldown were also not identified, and no information was provided about which snRNAs were pulled down. Thus, it is not clear if the OCRE domain can bind all or only some snRNPs containing SmB/B'/N. These questions are not mentioned in the present Discussion but dramatically affect any model for how OCRE alters spliceosome assembly. Given the nicely shown interaction of RBM5's OCRE domain with the proline-rich regions of SmB/B'/N, and the fact that these proteins are present in the U1, U2, U4 and U5 snRNPs, a more thorough analysis of the interaction partners of the OCRE domain in nuclear extract is needed. The pulldown analysis should be repeated, analyzing all proteins and RNAs that are bound by OCRE, as well as analyzing how the stringency of the buffers used during pulldown and wash affect the spectrum of interacting species.*

We have worked hard to determine the complement of U snRNAs associated to OCRE in our pull downs. Unfortunately, however, the results were inconclusive because a high background of U snRNAs was observed even in the control pull down samples. After many attempts to improve the signal-to-noise ratio using different pull down and stringency conditions, we were forced to accept that these assays do not provide evidence of consistent enrichment of any particular snRNP. One possibility is that the pull down somehow dislocates the snRNP-associated proteins from other RNP components, including the snRNAs. This is also consistent with the apparent absence of other Sm proteins from the precipitates. It seems likely that our pull-down analyses are a gross simplification of the sophisticated interactions that occur in the context of the spliceosome assembly pathway. Nevertheless, our assays revealed a direct interaction between the OCRE domain and the SmB/B'/N C terminal tail that, on the basis of our structure-function studies, is directly relevant for splicing regulation by RBM5.

We agree that an interaction with any U snRNP where the SmB/B’/N tails are accessible should be in principle possible. Looking at the recent high-resolution crystal and cryo-EM structures of spliceosomal complexes, we note that the long and flexible tails are predicted to be available for interactions in the tri-snRNP, in U1 snRNP and in complex B. We have expanded the text to discuss these structural insights and how contacts with either U1 / U2 or with the tri-snRNP could accommodate the activity of RBM5 as a repressor of the transition from complex A to B.

*4) Also in the previous study (Bonnal et al., 2008), addition of RBM5 to an in vitro splicing reaction led to the accumulation of A complexes, and a block to B complex formation on intron 5 or 6 of Fas pre-mRNA. This indicated that stable recruitment of the tri-snRNP was hindered in the presence of RBM5. What effect does a RBM5 OCRE domain alone, or an RBM5 mutant lacking OCRE or with an inactive OCRE domain have on splicing complex formation in vitro? This experiment would provide additional information about the mechanism whereby RBM5's OCRE domain affects splicing. For example, Rbm5 also interacts with U2AF. It is possible that this U2AF interaction is sufficient to block to tri-snRNP recruitment, and the OCRE/SmB interaction is serving other roles.*

We have tested the activity of the OCRE domain alone, as well as of a RBM5 mutant lacking the OCRE domain, in spliceosome assembly and in *in vivo* splicing assays. The results of the new Figure 6—figure supplements 1 and 2 show that the OCRE domain alone (in a construct that also includes the flanking KEKE domain) fails to inhibit spliceosome assembly in biochemical assays (Figure 6—figure supplement 1) and to regulate Fas exon 6 *in vivo* (Figure 6—figure supplement 2). In addition, new Figure 6 shows that both the deletion of the C-terminal region of RBM5 and a deletion of the OCRE domain compromise the ability of RBM5 to inhibit the transition from complex A to complex B in spliceosome assembly assays. Collectively, the results indicate that the OCRE domain is necessary but not sufficient to block B complex formation and that additional contributions from other parts in the C-terminal half of RBM5 contribute to this activity. One possibility, now discussed in the manuscript, is indeed that an interaction with U2AF, which has been previously mapped to a C-terminal region in RBM5, is important for regulation by RBM5. Investigating these additional interactions and their specific contributions to the modulation of spliceosome assembly will be the focus of our future work.